METHODS AND RESOURCES

# A *Drosophila* glial cell atlas reveals a mismatch between transcriptional and morphological diversity

Inês Lago-Baldaia[1], Maia Cooper[1]ᵒ, Austin Seroka[2]ᵒ¤, Chintan Trivedi[1], Gareth T. Powell[1], Stephen W. Wilson[1], Sarah D. Ackerman[3,4]*, Vilaiwan M. Fernandes[1]*

**1** Department of Cell and Developmental Biology, University College London, London, United Kingdom, **2** Institute of Neuroscience, Howard Hughes Medical Institute, University of Oregon, Eugene, Oregon, United States of America, **3** Department of Pathology and Immunology, Brain Immunology and Glia Center, Washington University School of Medicine, Saint Louis, Missouri, United States of America, **4** Department of Developmental Biology, Washington University School of Medicine, Saint Louis, Missouri, United States of America

ᵒ These authors contributed equally to this work.
¤ Current address: Basic Sciences Division, Fred Hutchinson Cancer Research Center, Seattle, Washington, United States of America.
* sarah.ackerman@wustl.edu (SDA); vilaiwan.fernandes@ucl.ac.uk (VMF)

**Data Availability Statement:** All raw and processed transcriptome data for the embryonic dataset are available from NCBI GEO (accession

## Abstract

Morphology is a defining feature of neuronal identity. Like neurons, glia display diverse morphologies, both across and within glial classes, but are also known to be morphologically plastic. Here, we explored the relationship between glial morphology and transcriptional signature using the *Drosophila* central nervous system (CNS), where glia are categorised into 5 main classes (outer and inner surface glia, cortex glia, ensheathing glia, and astrocytes), which show within-class morphological diversity. We analysed and validated single-cell RNA sequencing data of *Drosophila* glia in 2 well-characterised tissues from distinct developmental stages, containing distinct circuit types: the embryonic ventral nerve cord (VNC) (motor) and the adult optic lobes (sensory). Our analysis identified a new morphologically and transcriptionally distinct surface glial population in the VNC. However, many glial morphological categories could not be distinguished transcriptionally, and indeed, embryonic and adult astrocytes were transcriptionally analogous despite differences in developmental stage and circuit type. While we did detect extensive within-class transcriptomic diversity for optic lobe glia, this could be explained entirely by glial residence in the most superficial neuropil (lamina) and an associated enrichment for immune-related gene expression. In summary, we generated a single-cell transcriptomic atlas of glia in *Drosophila*, and our extensive in vivo validation revealed that glia exhibit more diversity at the morphological level than was detectable at the transcriptional level. This atlas will serve as a resource for the community to probe glial diversity and function.

GSE208324; https://www.ncbi.nlm.nih.gov/geo/query/acc.cgi?acc=GSE208324). The scripts used to process the raw RNA-seq data and extract neuronal and glial clusters from the embryonic dataset are available at https://github.com/AustinSeroka/2022_stage17_glia. All other scripts, including midline glia annotation from the whole embryonic dataset, cleaned-up and annotated embryonic glial dataset, as well as the integrated, cleaned-up and annotated young adult optic lobe glial dataset are available at https://github.com/VilFernandesLab/2022_DrosophilaGlialAtlas. We used published single cell RNA sequencing datasets of the optic lobes: Özel et al. (2021) (NCBI GEO accession GSE142787; https://www.ncbi.nlm.nih.gov/geo/query/acc.cgi?acc=GSE142787) and Kurmangaliyev et al. (2020) (NCBI GEO accession GSE156455; https://www.ncbi.nlm.nih.gov/geo/query/acc.cgi?acc=GSE156455). Numerical values used to generate graphs are included in S5 Data file. The cleaned-up and annotated embryonic glial dataset and the integrated cleaned-up and annotated young adult optic lobe glial dataset are included here as source data files: S6 and S7 Data files, respectively. All other relevant data are within the paper and its Supporting information files.

**Funding:** SWW was funded by a Wellcome Investigator Award (104682/Z/14/Z; https://wellcome.org). SDA was funded by the National Institute of Health (K99/R00NS121137; https://www.nih.gov). VMF was funded by a Wellcome Trust and the Royal Society Sir Henry Dale Research Fellowship (210472/Z/18/Z; https://wellcome.org and https://royalsociety.org). The funders had no role in study design, data collection and analysis, decision to publish, or preparation of the manuscript.

**Competing interests:** The authors have declared that no competing interests exist.

**Abbreviations:** APF, after puparium formation; BBB, blood–brain barrier; CNS, central nervous system; GO, Gene Ontology; HCR, hybridization chain reaction; HRP, Horseradish peroxidase; MCFO, multicolour flip-out; PBS, phosphate buffered saline; PCA, principal component analysis; RNA-seq, RNA-sequencing; TPM, transcripts per million; UMAP, uniform manifold approximation and projection; VNC, ventral nerve cord.

# Introduction

Nervous systems contain more distinct cell types than any other organ. This cellular diversity underlies the complexity and multifunctionality of circuits and processing networks in the brain and, thus, defines the breadth of an animal's behavioural repertoire. Not surprisingly, categorising neural cell types has long been, and continues to be, a major endeavour in the field. Although much emphasis has been placed on categorising neuronal diversity, we know much less about the extent of glial diversity. Given their pivotal roles in every aspect of nervous system development and function [1,2], understanding glial diversity is also imperative.

Morphological diversity among glia has been documented alongside that of neurons for over a century [3]. This morphological heterogeneity exists not only between broad glial classes (i.e., astrocytes, oligodendrocytes, Schwann cells, and microglia) but also within classes [4–7]. It has long been appreciated that mammalian astrocytes from different brain regions vary in morphology [3,8,9], and recent advances in RNA-sequencing (RNA-seq) technologies have revealed regionalized molecular diversity in astrocytes and other central nervous system (CNS) glial cell classes (e.g., oligodendrocyte progenitor cells and microglia) [10–16]. Confoundingly, astrocytes are known to be highly plastic cells. Most notably, in response to injury, astrocytes undergo a process called astrogliosis, become "reactive," and alter their morphology dramatically [17]. It is clear that astrocyte reactivity represents a change in cell state due to underlying differences in environment. Thus, in healthy conditions, it is difficult to distinguish whether the morphological diversity of astrocytes is a consequence of cell-fate diversity and/or cell-state. In other words, what is the relationship between glial morphology and transcriptional profile?

*Drosophila* glia share several key morphological and functional attributes with their vertebrate counterparts, including maintaining neurotransmitter and ionic homeostasis, providing trophic support for neurons, acting as immune cells, and modifying neural circuit function [1,18–20]. In the *Drosophila* CNS, neuropils contain synaptic connections, while neuronal cell bodies are located at the cortex, around the periphery of neuropils; axon tracts connect different neuropils to each other. *Drosophila* glia can be categorised based on morphology and by their association with these anatomical structures as either outer or inner surface glia, cortex glia, ensheathing glia, or astrocytes (Figs 1 and 2). Surface glia comprise 2 sheet-like glia called the perineurial and subperineurial glia [20,21]. Together, these form a double-layered surface that spans the nervous system, which acts as a blood (or hemolymph)–brain barrier (BBB) [20,21]. Cortex glia envelop neuronal cell bodies in cortical regions of the CNS, whereas ensheathing glia can wrap axonal tracts between neuropils (aka tract ensheathing glia) or wrap neuropil borders [20,22]. Astrocytes also inhabit neuropil regions with ensheathing glia, but extend many fine projections into the neuropil to associate with neuronal synapses, akin to vertebrate astrocytes [20,23]. Although *Drosophila* has a simplified nervous system with reduced numbers of glia relative to mammals, striking morphological diversity exists between and within glial cell classes during development and in the adult [22,24–26]. For example, in the highly ordered visual system, astrocytes of distinct morphologies can be found across neuropils and within the same neuropil [24,27]. Whether these morphological categories correspond to distinct subclasses with unique transcriptional profiles and functions is not known.

Here, we leverage the simple and tractable *Drosophila* nervous system to explore the relationship between glial morphological diversity and transcriptional signatures. We generated and validated a single-cell atlas of all glial classes in 2 distinct *Drosophila* circuits—the embryonic VNC and the adult optic lobe. We chose these circuits as they are well characterised with

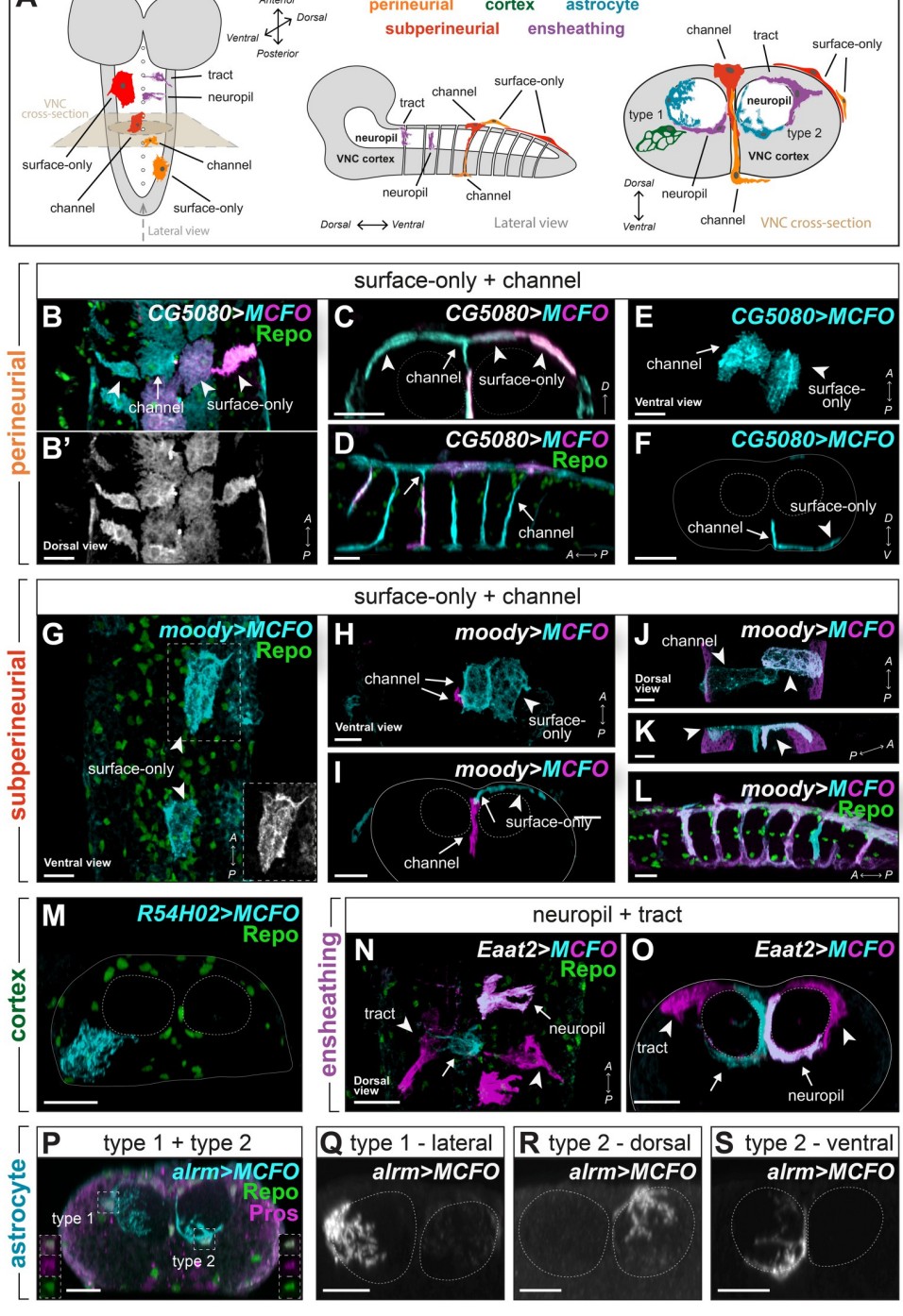

**Fig 1. Morphologies of newly hatched VNC glia. (A)** Schematics of the embryonic CNS along different axes with the 5 glial classes indicated. Cortex (neuronal cell bodies) indicated in grey. **(B)** Dorsal surface view of the VNC showing surface-only and channel-associated perineurial and subperineurial glial cells on the surface with characteristic fibrous membranes and loose tiling. **(C)** Cross-sectional view of the VNC showing the same cells as in (B) now showing the tapered projection from a (dorsal) channel-associated perineurial glial cell that tiles with neighbouring surface-only perineurial glial cells. **(D)** Lateral view of the VNC showing channel-associated perineurial glia on the ventral and dorsal surfaces, each sending a single projection with ventral channel-associated perineurial glia sending longer processes than their dorsal counterparts. **(E)** Ventral surface view of the VNC showing surface-only and channel-associated perineurial and subperineurial glial that tile with each other loosely. **(F)** Cross-sectional view of the VNC showing the same cells as in (C) now showing the tapered projection from a (dorsal) channel-associated perineurial

glial cell that tiles with a neighbouring surface-only perineurial glial cell. **(G)** Surface view of the VNC showing polygonal-shaped surface-only subperineurial glia on the surface. **(H)** Surface view of the VNC showing polygonal-shaped surface-only subperineurial glia (cyan) on the surface and an underlying channel-only subperineurial glial cell (magenta). **(I)** Cross-sectional view of the VNC showing the same cells as in (H). **(J, K)** Surface view (J) and oblique view (K) of the VNC showing 2 ventral surface- and channel-associated subperineurial glia (cyan and magenta) with extensions towards the neuropil. **(L)** Lateral view of the VNC showing subperineurial glia on the ventral and dorsal surfaces, sending projections along the channels forming tube-like structures. **(M)** Cross-sectional view of the VNC showing a single cortex glial cell in the cortical region forming a membranous, honeycomb-like structure. **(N, O)** Lateral view (N) and cross-sectional view (O) of the VNC showing ensheathing glial cells at the border of the cortex and neuropil. Tract ensheathing cells extend longer or shorter processes through axon tracts entering the neuropil, while neuropil ensheathing cells only extend processes along the neuropil border. **(P)** Cross-sectional view of the VNC showing a type 1 astrocyte and a type 2 astrocyte sending processes into the neuropil. Type 1 astrocyte processes were highly ramified, whereas type 2 astrocyte processes were less ramified. Co-expression of Pros and Repo indicated astrocyte identity (insets). **(Q–S)** Cross-sectional views of the VNC showing single astrocytes in greyscale, from dorsal, lateral, and ventral nucleus positions, with corresponding morphological type 1 or 2 indicated. Green marks Repo, and cyan and magenta mark MCFO clones in all panels, except (P) with Pros in magenta. Dashed lines outline the neuropil and full lines outline the VNC. Clones represent samples at 0 h after larval hatching. All scale bars represent 10 μm. CNS, central nervous system; MCFO, multicolour flip-out; VNC, ventral nerve cord.

many existing tools and reagents, they span 2 different circuit types (sensorimotor versus pure sensory), and they represent 2 distinct developmental stages. We identified several new glial morphological categories but found no clear correlation between glial morphological diversity and detectable transcriptional diversity. Moreover, we found that class-specific transcriptomes were conserved from embryo to adult, despite changes in circuit location and developmental stage. One exception was the glia of the optic lobe lamina, which accounted for the majority of glial diversity at a transcriptional level. The lamina and its associated glia lie in close proximity to an environmental interface, positioned immediately below photoreceptors of the compound eye; these glia were enriched for gene expression associated with immune-related functions. Our data suggest that within-class (e.g., astrocytes) glial morphological categories cannot be assumed to correspond to transcriptionally distinct subclasses. Instead, we propose that glia adopt different morphological and functional states in response to cues from their local environment.

## Results

### Morphological diversity of embryonic *Drosophila* glia

To determine the relationship between glial morphology and transcriptional signature, we began by characterising glial morphology across distinct *Drosophila* brain regions and developmental stages. We focused on glia in the VNC, akin to the vertebrate spinal cord, in the late embryonic (stage 17) CNS. The developing VNC contains 5 major glial classes: astrocytes, ensheathing glia, cortex glia, and 2 types of surface glia, which are all neuroectodermal in origin and express the marker *reversed polarity* (*repo*) [20]. We used enhancer trap Gal4 drivers expressed in each of these glial cell classes to sparsely label cells using the multicolour flip-out (MCFO) cassette. We then visualised single glial cell morphology at 0 hours after larval hatching (0h ALH) to assess morphological diversity both within and between classes (Fig 1). Note that in addition to the 5 major glial classes described above, the VNC contains a distinct class called the midline glia, which are a transient population found only during embryonic and larval stages [28–31]. Although midline glia express *wrapper*, otherwise known as a cortex glia marker [32–34], they do not resemble cortex glia in form or function but instead ensheath commissural axons and play critical roles in axon guidance and VNC morphogenesis [35]. Moreover, unlike the other major glial classes described above, midline glia are mesectodermal

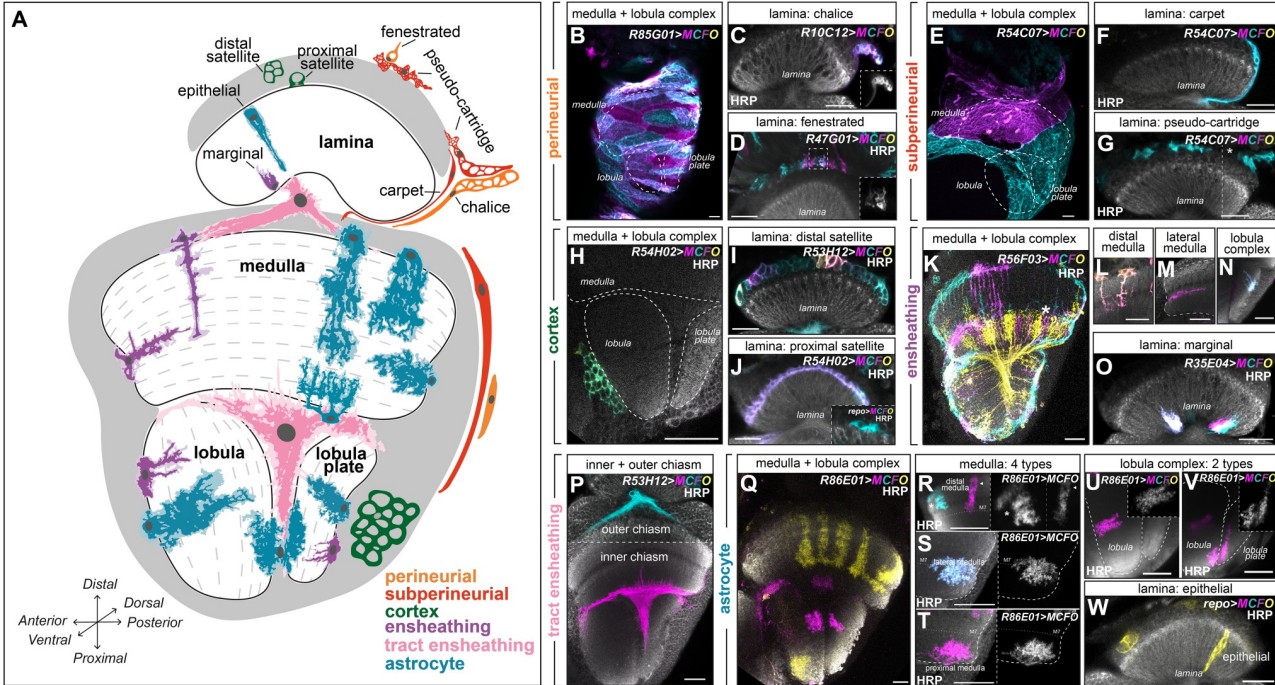

**Fig 2. Morphologies of glia in the adult optic lobe.** (A) Schematic of the cross-section of the adult optic lobe and its 4 neuropils: lamina, medulla, lobula, and lobula plate with glial classes indicated. Dashed lines indicate the layers of the specific neuropil. Cortex (neuronal cell bodies) indicated in grey. (B) Maximum projection showing MCFO clones of perineurial glial cells covering the medulla, lobula, and lobula plate. Cells were oblong-shaped and tiled together. (C) A cross-sectional view of the lamina showing a chalice glial cell (a type of lamina perineurial glia at the rim of the lamina cortex and neuropil). (D) A cross-sectional view of the lamina showing fenestrated glial cells (a type of lamina perineurial glia), separating the compound eye from the lamina. (E) Maximum projection showing MCFO clones of subperineurial glia that covered the medulla, lobula, and lobula plate as large squamous cells that tiled together. (F) A cross-sectional view of the lamina showing a single carpet glia (a type of lamina subperineurial glia), along the rim of the lamina cortex and neuropil. (G) A cross-sectional view of the lamina showing pseudocartridge glia (a type of lamina subperineurial glia), as irregularly shaped cells above the lamina cortex. (H) A cross-sectional view of the medulla, lobula, and lobula plate showing a single cortex glial cell in the lobula cortex with typical membranous and honeycomb-like morphology. (I) A cross-sectional view of the lamina showing distal satellite glia (a type of lamina cortex glia), excluded from the most proximal region of the lamina cortex. Note that inner chiasm glia were also labelled by this driver (bottom). (J) A cross-sectional view of the lamina showing proximal satellite glia in the most proximal region of the lamina. Inset shows a single proximal satellite glia labelled by *repo-Gal4*. (K) A cross-sectional view of the medulla, lobula, and lobula plate showing MCFO clones labelling ensheathing glia and a subset of neurons (asterisk). (L) Examples of ensheathing glia in the distal medulla sending primary projections from the surface of the neuropil to layer M7. (M) An example of an ensheathing glial cell in the lateral medulla sending a primary from the outer neuropil surface inwards along layer M7. (N) An example of an ensheathing glial cell in the lobula plate neuropil sending several short processes with minimal secondary branches into the neuropil. (O) A cross-sectional view of the lamina showing marginal glia (a type of lamina-specific ensheathing glia), which project partway into the lamina neuropil from the distal neuropil surface. (P) A cross-sectional view of the optic lobe showing 2 morphologically distinct tract ensheathing glia (called chiasm glia) located in the outer chiasm between the lamina and medulla, or the inner chiasm between the medulla, lobula, and lobula plate. Outer chiasm glia did not send projections into the neuropils, but the inner chiasm glia projected into all 3 neuropils. The dashed line separates images in the same brain on different z-stacks. (Q) A cross-sectional view of the medulla, lobula, and lobula plate showing astrocytes. (R) Examples of distal medulla astrocytes with short and long morphologies. The short distal medulla astrocyte (asterisk) projected to layer M6, while the long distal medulla astrocyte (arrowhead) projected to layer M8. (S) An example of a lateral medulla astrocyte projecting into the neuropil laterally along the M7/8 layers. (T) An example of a proximal medulla astrocyte (also called chandelier glia) projecting from the proximal surface of the medulla, up to layers M9 and M10. (U, V) Examples of astrocytes in the lobula and lobula plate, which (U) projected into the lobula neuropil from the cortex-neuropil border or (V) projected into both the lobula and pobula plate neuropils. (W) Example of an epithelial glial cell (right; the lamina astrocyte population), which projected across the entire distal-proximal neuropil length. Cyan, yellow, magenta, and purple mark the MCFO clones, while white labels HRP (labels the neuropils) in the main panels. Insets show the MCFO clones in greyscale. Dashed lines are used to outline neuropil borders or separate an inset. All scale bars represent 20 μm. HRP, Horseradish peroxidase; MCFO, multicolour flip-out.

in origin [36,37] and do not express the pan-glial marker *repo* or the broad glial marker *glial cells missing (gcm)* [35]. Midline glia have been characterised extensively by several groups [35,38–42]; therefore, given their distinct origin and the ambiguity surrounding their functional classification, we instead focused our analyses on *repo+* glia.

**VNC surface glia.** The *Drosophila* CNS is bathed in circulating hemolymph, which contacts the VNC at its main surface and along dorsoventral channels that perforate the VNC along the midline between pairs of longitudinal connectives and neighbouring neuromeres [43,44]. The surface glia comprise 2 glial classes, perineurial on the outer surface of the VNC and subperineurial glia positioned below perineurial glia [20,21]. Early glial studies did not distinguish between these 2 classes and referred to both collectively as subperineurial glia for their position below the perineurium [44,45]. An early characterisation of gene expression in glia of the embryo reported *CG5080* and *moody* expression in surface glia along the main VNC surface and associated with the dorsoventral channels ("channel glia"), a result since confirmed by bulk RNA profiling of surface glia [45,46]. It is important to note that these studies did not evaluate glial morphology, whether *CG5080* and *moody* were expressed by both surface glial classes, nor whether they were co-expressed in the same cells [45,46]. Indeed, Moody has since been shown to be expressed exclusively in subperineurial glia [47,48]. However, *CG5080* expression has not been characterised further, thus implying 3 possibilities: (i) it may be expressed in both perineurial and subperineurial glia; (ii) it may be expressed in perineurial glia exclusively; or (iii) it may be expressed in subperineurial glia exclusively.

To analyse surface glia morphology in the VNC, we used the *CG5080-Gal4* and *moody-Gal4* drivers to generate MCFO clones. *CG5080-Gal4* labelled cells with fibrous membranes and which tiled loosely with each other to cover the main surface of the VNC, typical characteristics of perineurial glia described by others [38] (Fig 1A–1F). Therefore, it is likely that *CG5080-Gal4* labelled cells that belong to the perineurial glial class. Interestingly, *CG5080-Gal4* labelled cells located along the midline of the dorsal and ventral surfaces of the VNC each sent a single tapered projection inward along the dorsoventral channels (Figs 1C, 1D, 1F, and S1A–S1C; $N$ = 51 clones from $N$ = 13 brains). When viewed at the surface, these cells tiled with and were indistinguishable from the other *CG5080-Gal4*-labelled surface-only cells (Fig 1B–1F; $N$ = 140 clones from $N$ = 13 brains). Ventral midline cells projected further than their dorsal counterparts and were present along the entire anterior-posterior axis of the VNC, whereas cells that projected from the dorsal surface were observed with lower frequency towards more posterior positions of the VNC (Figs 1D and S1A). To further confirm that the channel-associated cells belong to the perineurial glial class, we used *CG5080-Gal4* to label individual cells while co-labelling all glia with *repo-LexA>LexAop-myr::tdTomato*. We observed that the channel-associated cells occupied the outermost glial surface of the VNC (S1B and S1C Fig). Taken together with the fact that these cells tiled with surface-only perineurial glia, these data argue that they are a subclass of perineurial glia. Hereafter, we refer to them as "channel-associated perineurial glia."

Since *moody* is a known marker of subperineurial glia [47,48], we next generated MCFO clones labelled with *moody-Gal4*. Although we recovered cells at the surface of the VNC and associated with the dorsoventral channels, these did not resemble clones labelled by *CG5080-Gal4* (Fig 1A–1F). Instead, consistent with previous reports of subperineurial glia [38], *moody-Gal4* MCFO clones at the surface of the VNC were polygonal in shape, tiled tightly with each other, were larger than *CG5080-Gal4*-labelled cells, and displayed a honeycomb-like pattern within their cell boundaries, as a result of their membranes cupping neuronal cell bodies at the outer surface of the cortex [25] (Figs 1A, 1G–1L, and S1D). While most cells labelled by *moody-Gal4* were only associated with the main surface of the VNC (69.3% of total clones were associated with the main surface only; $N$ = 101 clones from $N$ = 12 brains), some of the cells, present along the VNC midline sent short (non-tapering) projections along the dorsoventral channels (21.8% of total clones). In addition, *moody-Gal4* also labelled cells that exclusively lined the dorsoventral channels (8.9% of total clones; Fig 1H and 1I). Thus, we

observed a continuum of glial morphologies between the 2 extremes of surface-only and channel-only subperineurial glia (Fig 1J–1L). Importantly, glia with intermediate morphologies (i.e., which associated with both the surface and the channels) tiled tightly with neighbouring surface-only associated cells (Fig 1H and 1I).

As aforementioned, *CG5080* and *moody* were reported to be expressed in surface glia though co-expression and expression across both surface glial classes were not assessed [45,46]. Here, we used *CG5080-Gal4* and *moody-Gal4*-labelled MCFO clones to evaluate cell morphology, tiling properties, size and position, and clarify whether these labelled the same or different surface glial classes. Consistent with previous reports [38,47,48], our data argue that *moody-Gal4* labels the subperineurial glial class, which can be further subdivided into surface-only, surface- and channel-associated, and channel-only morphological categories. By contrast, our data suggest that *CG5080-Gal4* labels the perineurial glial class, which can be further subdivided into surface-only and channel-associated morphological categories. *CG5080-Gal4*-labelled cells did not resemble cells labelled by *moody-Gal4*. Instead, they covered smaller domains at the outermost surface of the brain, possessed fibrous membranes and tiled loosely with each other, all features that are associated with perineurial glia [38]. Thus, we confirm *moody* as a marker of subperineurial glia and identify *CG5080* as a new marker of perineurial glia. Moreover, we show that surface glia associated with the dorsoventral channels (referred to as "channel glia" in early reports) [44,45] also consist of perineurial and subperineurial glia, which had not been appreciated previously.

**VNC cortex glia, ensheathing glia, and astrocytes.** *Drosophila* have 3 glial cell classes that collectively perform the functions of vertebrate astrocytes: cortex glia, ensheathing glia, and astrocytes [20]. To analyse their morphology, we used the *R54H02(wrapper fragment)-Gal4*, *Eaat2-Gal4*, and *alrm-Gal4* drivers to generate MCFO clones, respectively. In the VNC, cortex glia (*wrapper*+) varied in size but exhibited 1 general morphology. In brief, all cortex glia extended a large membrane sheet to envelop neighbouring neuronal cell bodies. Furthermore, each cortex glia extended processes to contact the apical and basal surfaces of the cortex, which were occupied by other glial cell membranes (Fig 1A and 1M; *N* = 295 clones from *N* = 13 brains) [20].

Two morphological categories of VNC ensheathing glia have been described previously: (i) ensheathing glia associated with the ventral and medial neuropil, which wrap the border between the cortex and neuropil with no projections extending into the neuropil itself; hereafter referred to as neuropil ensheathing glia; and (ii) ensheathing glia associated with the dorsal neuropil and the proximal regions of intersegmental nerves (the latter are also known as "ensheathing/wrapping glia" or "tract ensheathing glia") [25,26]. We recovered both morphological categories of ensheathing glia with *Eaat2-Gal4* (79.6% of clones were neuropil-only ensheathing glia and 20.4% of clones were tract ensheathing glia; Fig 1A, 1N, and 1O). We note that tract ensheathing glia varied in the degree to which they associated with the neuropil and axon tracts (see examples in Fig 1N and 1O; *N* = 190 clones from *N* = 11 brains).

Using *alrm-Gal4*, we observed 2 morphological categories when we labelled astrocytes (*alrm*+) by MCFO (Fig 1P–1S), which we named type 1 and type 2 (Fig 1A and 1P–1S; *N* = 117 clones from 36 brains). Type 1 astrocytes extended highly ramified processes throughout the neuropil (Fig 1A, 1P, and 1Q) and represented 37 ± 6 SEM% of clones. Type 2 astrocytes elaborated some processes within the neuropil, but also extended a single radial process along the border of the neuropil and cortex (Fig 1A, 1P, 1R, and 1S) and represented 62 ± 4 SEM% of clones. Type 2 astrocytes were distinguished from ensheathing glia by nuclear expression of Prospero (Fig 1P) [26]. The VNC contains a single neuropil regionalised into dorsal motor and ventral sensory processing domains [49–53]. We wondered whether type 1

and type 2 astrocytes displayed regional associations that might indicate morphological specialisation to distinct circuit types. Indeed, it was reported previously that astrocyte cell bodies could be consistently allocated to one of 3 distinct regions around the neuropil: dorsal, lateral, and ventral and that the neuropil domains covered by astrocytes from these regions were also stereotyped corresponding to either the dorsal-medial, lateral, or ventral-medial domains [26]. To probe for astrocytic morphological specialisation between the dorsal (motor) and ventral (sensory) processing domains of the neuropil, we used astrocyte cell body position along the neuropil to allocate them to either the dorsal, lateral, or ventral domains [26] (S2A Fig and S1–S3 Videos; $N$ = 117 clones from 36 brains) and correlated these positions to type 1 or type 2 morphologies (S2B Fig). We found that astrocytes with cell body positions along the dorsal (motor) and ventral (sensory) neuropil border were predominantly type 2, whereas laterally positioned astrocytes were predominantly type 1 (lateral versus dorsal and lateral versus ventral, $p < 0.0001$, dorsal versus ventral, $p = 0.593$; Fisher's exact test). Thus, type 1 and type 2 astrocytes show regional preferences, but these did not correlate with sensory versus motor circuit types. To further quantify their morphological differences, we measured total cell volume and the number of primary branches from the cell body (S2C and S2D Fig). Unsurprisingly, lateral astrocytes (more ramified type 1 morphology) occupied larger volumes with more primary branches on average compared to either dorsal or ventral astrocytes (type 2 morphology) (total volume: lateral versus dorsal. $P = 0.0002$ and lateral versus ventral, $p = 0.004$; number of branches: lateral versus dorsal. $P = 0.0005$ and lateral versus ventral, $p = 0.0097$; Mann–Whitney U-test; S2C and S2D Fig). However, dorsal and ventral astrocytes did not differ from each other in their cell volumes or the number of primary branches (total volume, $p = 0.101$; number of branches: $p = 0.398$; Mann–Whitney U-test; S2C and S2D Fig).

In summary, within the early larval VNC, cortex could be defined by a single stereotyped morphology, whereas perineurial, subperineurial, ensheathing, and astrocyte glial classes each contained morphologically distinct subpopulations (Fig 1 and S1 File).

## Optic lobe glia display morphological diversity within glial classes

In addition to the relatively simple VNC, we also focused on the adult optic lobe, which is more structurally complex than the VNC with 4 distinct neuropils called the lamina, medulla, lobula, and lobula plate (Fig 2A). Several other groups have characterised the morphology of each of the 5 major glial classes (perineurial, subperineurial, cortex, ensheathing, and astrocyte) present in the optic lobe in detail [24,25,27]. Therefore, we used previously characterised Gal4 lines [24,25,54] (S3 Fig) to generate MCFO clones to visualise glial morphologies and validate morphological diversity within each class (summarised in Fig 2; see Materials and methods for more details).

**Optic lobe surface glia.** Briefly, we used *R85G01-Gal4* to label perineurial glia that covered the medulla, lobula, and lobula plate neuropils and observed morphological homogeneity across these 3 neuropils (Fig 2A and 2B) [25]. We used *R10C12-Gal4* to label chalice glia, a putative perineurial glial population found at the margins of the lamina neuropil (Fig 2A and 2C). Finally, we used *R47G01-Gal4* to label fenestrated glia, specialised perineurial glia found over the lamina cortex (Fig 2A and 2D) [25]. We then used *R54C07-Gal4* to label subperineurial glia of the optic lobe [25]. Subperineurial glia of the same morphology covered the medulla, lobula, and lobula plate neuropils (Fig 2A and 2E). In the lamina, *R54C07-Gal4* also labelled carpet glia, a specialised subperineurial glia found at the margins of the lamina neuropil [24,25] and pseudo-cartridge glia, a specialised subperineurial glia adjacent to the lamina cortex (Fig 2A, 2F, and 2G). Thus, we validated each of the previously annotated surface glial subtypes [24,25].

**Optic lobe cortex glia, ensheathing glia, and astrocytes.** To label optic lobe cortex glia, we used *R54H02-Gal4* (Gal4 driven by a fragment of the *wrapper* locus) and *R53H12-Gal4* [25]. *R54H02-Gal4*-labelled cortex glia in the medulla, lobula, and lobula plate, which were morphologically indistinguishable across the 3 neuropils, and the proximal satellite glia, a lamina-specific cortex glia located in the proximal lamina (Fig 2A, 2H, and 2J) [25]. *53H12-Gal4* labelled the distal satellite glia, another lamina-specific cortex glia population, located in the distal lamina (Fig 2A and 2J) [25].

To visualise optic lobe ensheathing glia, we used *R56F03-Gal4*, *R35E04-Gal4*, and *R53H12-Gal4* [25]. *R56F03-Gal4* labelled lamina, medulla, lobula, and lobula plate ensheathing glia (S3F Fig) [25]. The medulla exhibited 2 ensheathing glial morphologies: one in the distal medulla, which sent a primary process to layer M7 and one in the lateral medulla, which sent a primary process along layer M7 (Fig 2A and 2K–2M) [25]. Ensheathing glia of the lobula and lobula plate showed more complex branching patterns compared to the medulla and were less stereotyped (Fig 2A, 2K, and 2N). *R56F03-Gal4* also labelled the marginal glia, a lamina-specific ensheathing glia (S3F Fig) [25], which we labelled specifically by *R35E04-Gal4* (Figs 2A, 2O, and S3J). Finally, *R53H12-Gal4*-labelled chiasm glia, a specialised ensheathing glia (tract ensheathing glia), which ensheaths tracts of neuronal projections between neuropils. Chiasm glia displayed 2 distinct morphologies: one associated with the chiasm between the lamina and medulla (outer chiasm) and the other associated with the chiasm between the medulla, lobula, and lobula plate chiasm (inner chiasm) (Fig 2A and 2P) [24,25].

Finally, to label optic lobe astrocytes, we used *R86E01-Gal4* [25]. In the medulla, we observed 4 distinct astrocyte morphologies as previously reported: 2 in the distal medulla, a third in the lateral medulla, and a fourth in the proximal medulla (also called chandelier glia) (Fig 2A and 2Q–2T) [24,25,27]. In addition, the lobula and lobula plate were together populated by 3 morphologically distinct astrocyte populations (Fig 2A, 2Q, 2U, and 2V). Finally, *R86E01*-Gal4 also labelled astrocytes of the lamina, called epithelial glia [25] (shown instead with *repo-Gal4*) (Fig 2A and 2W).

Since astrocytes displayed the most within-class morphological diversity of all the glial classes in the optic lobe, often with multiple stereotyped morphologies occupying the same neuropil, we sought to characterise them further. The medulla, lobula, and lobula plate neuropils display synaptic stratifications (layers), which arise because of diverse neuronal arborization patterns [55]. Different neurons project to different layers and in this way restrict the partners with whom they synapse in a layer-specific manner. Thus, distinct layers encode distinct features of visual information [55]. We found that astrocytes which differed dramatically in shape sometimes had similar whole cell volumes and similar numbers of primary branches (S4A–S4C Fig). Therefore, to better assess whether astrocyte morphological categories displayed layer-specific and, therefore, circuit-specific associations, we quantified their volumes across neuropil layers (S4D and S4E Fig). This analysis revealed that each astrocyte morphological category displayed a clear preference in the layers it covered (S4D and S4E Fig and S3–S11 Videos).

In sum, all major classes of optic lobe glia showed some regionalized morphological differences (Fig 2A and S2 File). Additionally, for some glial classes—cortex glia in the lamina, ensheathing glia in the medulla, and astrocytes in the medulla, lobula, and lobula plate—we observed morphologically distinct populations in close proximity to each other within the same neuropil. Thus, optic lobe glia exhibited much more morphological diversity than VNC glia, prompting us to ask how transcriptional diversity differs between glia at embryonic and adult stages. Furthermore, are distinct glial morphologies within broad glial classes associated with unique transcriptional signatures?

## A transcriptomic atlas of embryonic and young adult *Drosophila* glia

To correlate glial cell morphology and transcriptional identity during development, we performed scRNA-seq on late-stage *Drosophila* embryos (stage 17). Whole embryos were dissociated into a single cell suspension, filtered, prepared using the 10X Genomics single cell pipeline, and sequenced via Illumina sequencing (see Materials and methods and S5 Fig). At this stage and through larval stage 1, we quantified an average of 528 ± 9 SEM glial cells in the CNS (labelled by the glial-specific marker Reversed polarity or Repo) that were localised in the VNC, and 188 ± 9 SEM glial cells in the brain lobes (*N* = 8 brains per anatomical region). Thus, our dataset was enriched for VNC glia, which make up 74% of the CNS glia at this time (*p* < 0.0001, one-way ANOVA). Following sequencing, cells were clustered based on differential gene expression using Seurat, and glial clusters were computationally isolated by expression of the pan-glial markers *repo* and the astrocyte-specific markers *GABA transporter (gat)* and *astrocytic leucine-rich repeat molecule (alrm)*, except for the midline glia, which lack *repo* expression [35] (see Materials and methods and Fig 3 for further details). We bioinformatically isolated midline glia based on *wrapper* and *single minded (sim)* expression [34,56] and performed hierarchical cluster analysis on midline glia, all *repo*+ glial clusters and neuronal clusters (see Materials and methods and S6A–S6F Fig for more details). This analysis revealed that midline glia formed an outgroup to neuronal and *repo*+ glial clusters, perhaps not surprisingly given their distinct (mesectodermal) origin [36,37] (see S1 Data for genes with enriched expression in midline glia). We therefore continued all further analyses focusing on *repo*+ glia only but nonetheless validated *wrapper* expression in midline glia in vivo using *wrapper-Gal4* (carried on a BAC insertion; [33]) (S6G and S6H Fig).

For the optic lobes, we used data from 2 recent studies that performed scRNA-seq on whole optic lobe tissue at the young adult stage and throughout pupal development [57,58]. Despite slight differences in experimental approaches between these studies (see Fig 3 for summary), both analyses generated 19 glial clusters for the young adult optic lobes (S7A and S7B Fig); however, since both studies focused on neuronal development, glial clusters were not annotated or analysed [57,58]. Since glial cells constitute only 10% to 15% of neural cells in the *Drosophila* nervous system [25], we sought to increase the number of cells analysed to maximise our ability to identify rare cell types. Therefore, we isolated glial cells based on their original annotation in each study (i.e., *repo* expression) and then combined the closest developmental stages using the Seurat integration pipeline to remove batch effects between libraries (Fig 3 and Materials and methods). Here, we focused primarily on the 3-day-old adult dataset from (7,544 cells) [57] integrated with the 96 h after puparium formation (APF) dataset from (766 cells) [58]; hereafter referred to as the young adult dataset (Figs 3, S7C, and S7D).

On uniform manifold approximation and projection (UMAP) visualisations, we noticed that many clusters were connected by streams of cells, which expressed *Resistant to dieldrin (Rdl)*, *Frequenin 1 (Frq1)*, and *Nckx30C* (S8A Fig, top). These streams also co-expressed *embryonic lethal abnormal vision (elav)* and *found in neurons (fne)*, whose expression is known to be enriched pan-neuronally (S8A Fig, bottom). Indeed, on UMAP visualisations *Rdl*, *Frq1*, and *Nckx30C* were expressed pan-neuronally (S8B Fig). We also detected *elav*, *fne*, *Rdl*, *Frq1*, and *Nckx30C* co-expressed in a subset of the embryonic dataset (S8C and S8D Fig). These data suggested that the *Rdl*, *Frq1*, and *Nckx30C* expressing glial cells could represent neuronal contamination. Therefore, we used in situ hybridization chain reaction (HCR) and MCFO clonal analyses to assess the expression of these genes in adult optic lobes and newly hatched larval VNCs, respectively, but failed to detect any expression in glial cells (S9A–S9D Fig). As glia interact closely with neurons throughout life, we hypothesised that the cells in our datasets that co-express *Rdl*, *Frq1*, and *Nckx30C* represented either glial cells contaminated by neuronal

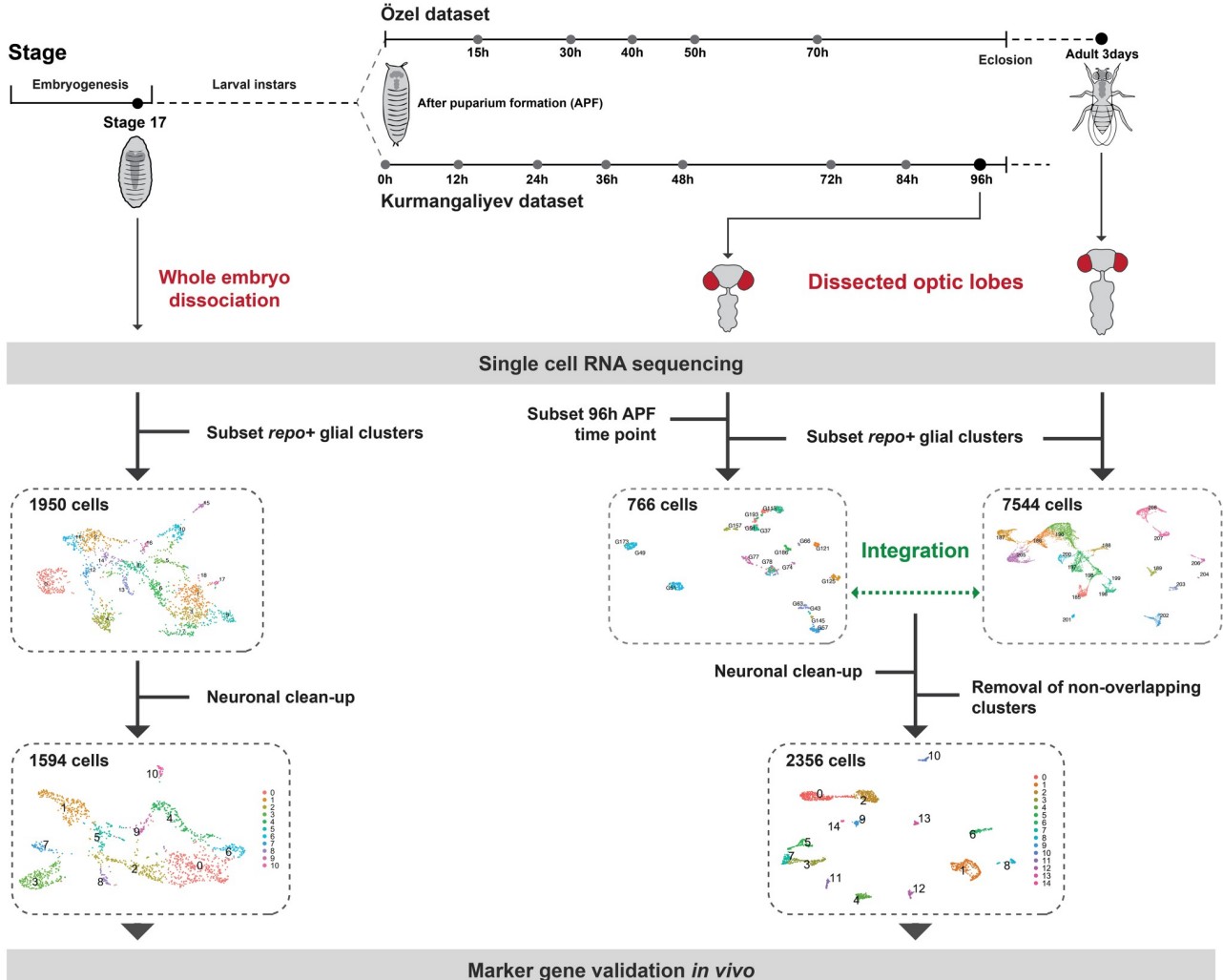

**Fig 3. Summary of the experimental and computational workflow used in this study. (Left)** We performed scRNA-seq on whole stage 17 embryos. We then subsetted *repo+* glial cells and eliminated neuronal contamination (see Materials and methods). This resulted in 1,594 cells and 11 clusters, which we proceeded to annotate by validating marker gene expression in vivo. **(Right)** We isolated and integrated *repo+* glial clusters from 2 published datasets of dissected optic lobes [57,58] (see also S6 Fig). We eliminated neuronal contamination and clusters that contained cells from only the Özel dataset (see Materials and methods) (see also S7 and S8 Figs). This resulted in 2,356 cells and 15 glial clusters, which we went on to annotate by comparisons with cell type-specific bulk RNA sequencing datasets and by validating marker gene expression in vivo. The data underlying this figure can be found at NCBI GEO accessions GSE208324, GSE142787 and GSE156455, and GitHub: https://github.com/AustinSeroka/2022_stage17_glia and https://github.com/VilFernandesLab/2022_DrosophilaGlialAtlas. APF, after puparium formation.

transcripts or neuronal cells contaminated by glial transcripts. Therefore, to circumvent potential clustering artefacts, we removed cells expressing high levels of *Rdl*, *Frq1*, and *Nckx30C* (see Materials and methods; S9E–S9H and S9J–S9N Fig). In addition, we noticed that a few cells expressed high levels of *Hemolectin* (*Hml*), a hemocyte-specific marker [59], which likely indicated contamination from a few stray hemocytes; therefore, we eliminated these cells also (see Materials and methods). Lastly, others have reported that glial clustering is sensitive to batch effects [57,58,60], and to further minimise these, we eliminated clusters to which the Kurmangaliyev dataset contributed fewer than 1% of the total number of cells in the cluster (S9I Fig). Following this elimination and reclustering (see Materials and methods), our datasets clustered

into 11 embryonic glial cell clusters and 15 adult optic lobe glial clusters (Figs 3, S9N, and S10A).

## Annotating embryonic glial clusters by validating marker gene expression in vivo

Based on known marker genes, we were able to make predictions about the identity of most of the embryonic clusters (Fig 4A and 4B). For example, cells in cluster 1 strongly expressed the gene *wrapper*, which other groups have previously reported as a cortex glia-specific marker with *wrapper* expression validated using a Gal4 driven by an intronic fragment from the *wrapper* locus (*R54H02-Gal4*) [61–64]. To validate the identity of each embryonic glial cluster, we acquired Gal4 lines for marker genes that were significantly enriched in one or more clusters (Fig 4A and 4B, see Materials and methods for complete list) and used these lines to generate MCFO clones (Figs 4C–4H and S11–S13). Embryos were heatshocked between 6 and 10 h after egg laying (prior to gliogenesis [65]), and larvae were dissected at 0 h after larval hatching. These MCFO analyses revealed near perfect specificity for the predicted glial cell type (Figs 4B and S11–S13). Indeed, by morphology and marker gene expression, 100% of *CG6126-Gal4* MCFO clones and 17.9 ± 1.8 SEM% of *pippin-Gal4* MCFO clones were perineurial glia; 100% of *CG10702-Gal4* MCFO clones, 80 ± 8.9 SEM% *Ntan1-Gal4* MCFO clones, and 100% of *moody-Gal4* MCFO clones were exclusive to subperineurial glia. We note that similar to a previous report [66], *pippin* is also expressed in subperineurial glia at low levels (Fig 4B), though we did not recover any subperineurial clones with the *pippin-Gal4* driver (S11I Fig; see below for further resolution of perineurial and subperineurial glial clusters). Furthermore, 98 ± 1 SEM% of *Eaat2-Gal4* MCFO clones were ensheathing glia, 96 ± 1.3 SEM% of *R54H02(wrapper fragment)-Gal4* MCFO clones were cortex glia, and 98 ± 1.5 SEM% of *alrm-Gal4* MCFO clones were astrocytes, consistent with their expression in the scRNA-seq data (Figs 4 and S11–S13). Importantly, drivers inserted in genes that showed expression across multiple clusters always produced MCFO clones with morphologies that matched the predicted cluster identities. In other words, a gene that showed expression in both the predicted astrocyte and ensheathing glial clusters produced both astrocyte and ensheathing glia clones (S11–S13 Figs). In this way, we annotated 6 of the 11 clusters as surface-only perineurial glia, channel-associated perineurial glia, subperineurial glia (all morphological categories), cortex glia, ensheathing glia (including both neuropil ensheathing and tract ensheathing), and astrocytes and uncovered novel marker genes for the major glial classes (and subclasses; summarised in Fig 4B). *Lobe (L)*, a known marker of peripheral nervous system glia [67], was expressed by 3 of the remaining unannotated clusters (clusters #0, #6, and #8). We therefore annotated these clusters as peripheral nervous system glia (PNSg_1, 2, and 3) but did not validate markers for these in vivo. In this way, we extended previous single-cell atlases, which included glial cells of the larval CNS and adult VNC [62,64], by validating marker gene expression in vivo and resolving surface glial classes into perineurial (surface-only and channel-associated) and subperineurial classes, which were indistinguishable in previous datasets.

## Perineurial glia morphologies were transcriptionally distinct

Our previous clonal analysis unveiled 2 perineurial glia morphologies: surface-only perineurial glia and channel-associated perineurial glia (Figs 1A–1F and S1). Validating cluster marker expression in vivo revealed that surface-only and channel-associated perineurial glia were transcriptionally distinct, with cluster #4 corresponding to surface-only perineurial glia and cluster #10 corresponding to channel-associated perineurial glia. Clusters #4 and #10 were located adjacent to each other on the UMAP. *PRL-1* and *pippin* were enriched in cluster #4

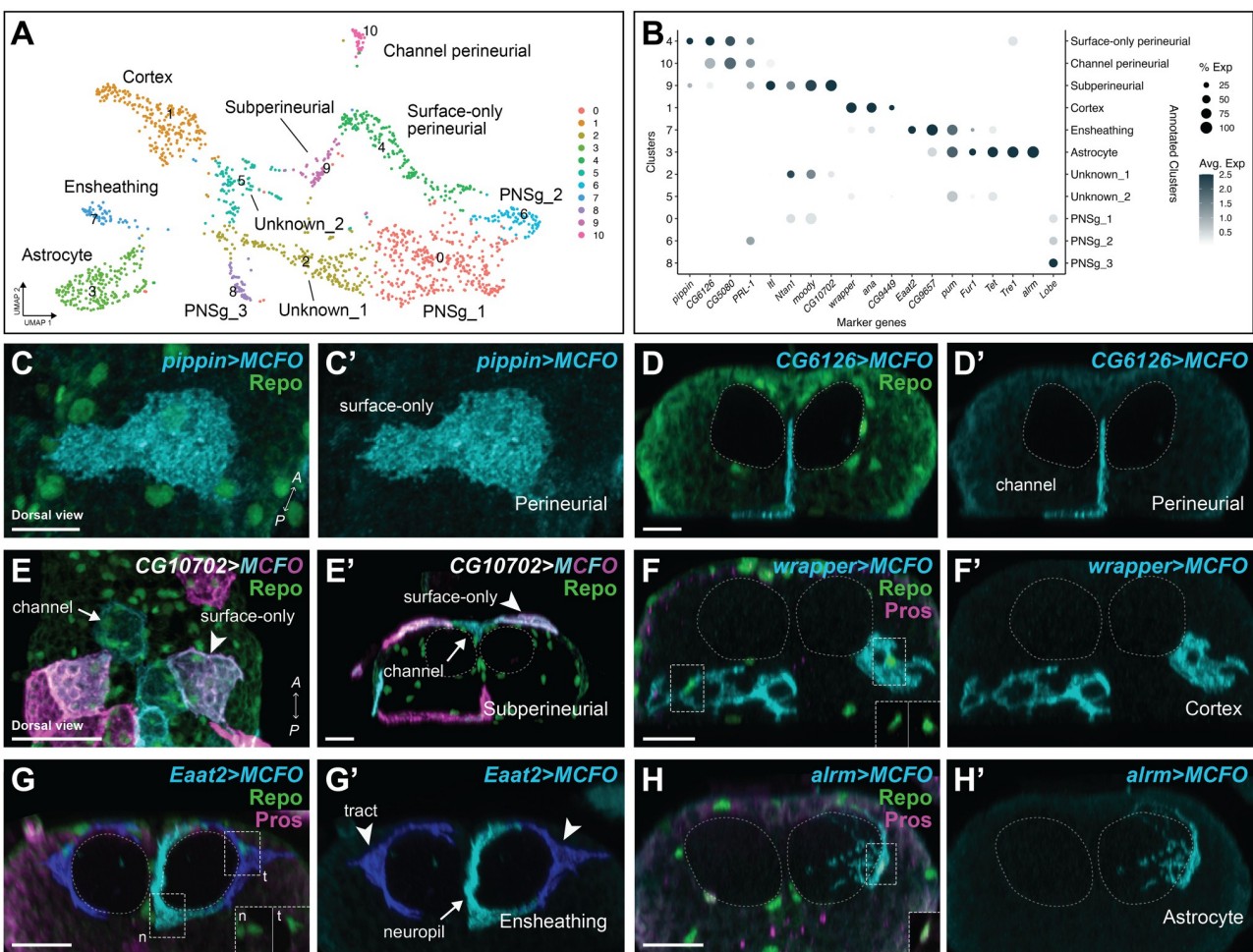

**Fig 4. Annotation of embryonic glial clusters in the VNC. (A)** UMAP of the 11 embryonic glial clusters labelled with both the cluster number (left) and our annotation of the specific glial class (and subtype; right) based on marker gene validation. **(B)** Expression plot of all marker genes selected for annotation, which were validated in vivo (except for *Lobe)*. The size of the dot represents the percentage of cells with expression in each cluster, while the colour of the dot represents the level of average expression in the cluster. **(C–H)** MCFO clones (0 h after larval hatching) generated with the Gal4 lines indicated belonged to the main embryonic glial subtypes as accessed by their morphology. **(C)** Surface-only perineurial ($N$ = 269 clones from $N$ = 11 brains); **(D)** the newly defined perineurial glial subtype, termed channel-associated perineurial glia ($N$ = 51 clones from $N$ = 13 brains); **(E)** subperineurial, all morphological categories ($N$ = 95 clones from $N$ = 12 brains); **(F)** cortex ($N$ = 295 clones from $N$ = 13 brains); **(G)** ensheathing, both tract (t) and neuropil (n) ($N$ = 190 clones from $N$ = 11 brains); and **(H)** astrocyte ($N$ = 117 clones from $N$ = 36 brains). All clones labelled in cyan, with Repo in green and Prospero in magenta. Insets in (F–H) show Prospero and Repo in glial nuclei in *alrm>MCFO* clones in (H) were positive for Prospero. See S11–S13 Figs for additional in vivo marker gene validation. Dashed lines outline the neuropils. Prime panels show MCFO clones alone. Scale bars are 10 μm. The data underlying (A, B) can be found in S2 and S6 Data files. MCFO, multicolour flip-out; VNC, ventral nerve cord; UMAP, uniform manifold approximation and projection.

(Fig 4B) and Gal4 drivers for these markers predominantly labelled clones with surface-only perineurial glial morphology (84.6% of 16 brains contained exclusively surface-only perineurial glia clones labelled by *PRL-1-Gal4*; 87.5% of 13 brains contained exclusively surface-only perineurial glia clones labelled by *pippin-Gal4*; Figs 4B, 4C, S11C, S11D, and S11J). By contrast, cluster #10 expressed high levels of *CG6126* and *CG5080*, but low expression of *PRL-1* and *pippin* (Fig 4B). Consistent with these expression patterns, 100% of brains ($N$ = 11 brains) contained both surface-only and channel-associated perineurial glia clones labelled by *CG6126-Gal4*, and 92.9% of brains ($N$ = 13 brains) contained both surface-only and channel-

associated perineurial glia clones labelled by *CG5080-Gal4* (Figs 1B–1F, 4D, and S11J). Thus, the 2 perineurial glia morphologies are transcriptionally distinct.

## Transcriptional profiles for subperineurial glia, ensheathing glia, and astrocytes did not distinguish morphological categories

Our prior clonal analyses revealed morphological heterogeneity in the subperineurial glia population, along a continuum from surface-only to channel-only (Fig 1H–1L), as well as 2 morphological categories within ensheathing glia: neuropil-only and tract-associated (Fig 1N and 1O), and 2 morphological categories within astrocytes: type 1 and type 2 (Fig 1P–1S). Interestingly, while our annotations revealed separate transcriptional clusters corresponding to surface-only perineurial glia and channel-associated perineurial glia (Figs 4A–4D and S11), we could resolve just 1 transcriptional cluster for subperineurial glia based on strong expression of *CG10702*, *Ntan1*, and *moody* (cluster# 9; Fig 4B and 4E); 1 transcriptional cluster for ensheathing glia based on strong expression of *Eaat2* (cluster #7; Figs 1N, 1O, 4B, and 4G); and 1 transcriptional cluster for astrocytes based on strong expression of *alrm* (cluster #3; Figs 1P–1S and 4B). We questioned whether any of the markers enriched in our astrocyte cluster might distinguish type 1 from type 2 astrocytes. To this end, we generated MCFO clones under the control of enhancers for genes expressed in the astrocyte cluster. All drivers gave rise to clones containing both type 1 and type 2 astrocytes at the expected proportions based on our *alrm* MCFO study (approximately 38% type 1 and approximately 62% type 2), with the exception of *pum*, which gave a slightly higher proportion of type 1 astrocytes (52%), but still contained both clones (Figs 4A, 4B, 4H, S12, and S13). Similarly, we failed to identify any markers that distinguished the subperineurial glial morphologies or the ensheathing glial morphologies (Figs 4E, 4G, S11, and S13). Thus, although distinct perineurial glia morphologies corresponded to distinct transcriptional signatures, the same was not true for subperineurial, ensheathing, or astrocyte morphologies, with the caveat that scRNA-seq may fail to detect genes that are expressed at low levels.

Overall, these data suggest that morphological diversity cannot be equated with transcriptional diversity, at least at present levels of detection. As the developing *Drosophila* VNC contains a single, simple, neuropil, which may not accurately represent the diversity of more complex brain regions, we next turned to the more complex adult *Drosophila* optic lobe to assess how accurately cellular identity can be gauged by glial morphology.

## Annotating young adult optic lobe glial clusters

**Comparisons to glial cell type-specific bulk RNA sequencing datasets.** To annotate adult optic lobe glia, we compared glial cell-type specific transcriptomes (obtained from bulk-RNA sequencing of FACS-purified glial types), which were published for the proximal satellite, epithelial, and marginal glia [68], to the integrated young adult optic lobe scRNA-seq dataset. This approach matched the proximal satellite glia with cluster #14 (Pearson correlation = 0.243; S10A–S10C Fig); however, both the epithelial and marginal glia showed the highest Pearson correlation with cluster #9 (0.277 and 0.3, respectively; S10A, S10D, S10E Fig), a small cluster made up of only 82 cells. Since the clustering algorithm we used has a known tendency to group together rare cell types while splitting apart abundant cell types artificially, we hypothesised that clusters containing few cells may contain more than 1 rare cell type [60,69]. To determine if cluster #9 was comprised of a mixture of cells belonging to epithelial and marginal glial cell types that were artificially merged because of their rarity, we analysed cluster #9 in isolation and found that it could be divided into 2 distinct subclusters (S10F Fig). We identified 22 genes that were differentially expressed (by at least 4-fold) between the 2 subclusters (S14 Fig).

We found that marginal- and epithelial-specific marker genes, identified from the FACS-purified transcriptomes, segregated perfectly between the 2 subclusters (e.g., *GstT4* for marginal glia and *CG43795* for epithelial glia; S10G and S10H Fig). Indeed, when we plotted the expression of these marker genes on the original UMAP, we observed a clear spatial segregation among the cells of cluster #9, supporting the hypothesis that cluster #9 contained a heterogeneous cell population made up of both epithelial and marginal glial cells (S10I and S10J Fig). Therefore, we used the subclusters to manually divide cluster #9 into 2 clusters (renamed cluster #9 and cluster #16), with (new) cluster #9 likely corresponding to the epithelial glia and cluster #16 likely corresponding to the marginal glia (S10K Fig).

To test for heterogeneity in other clusters, we subclustered each in isolation and examined the differential gene expression between subclusters. To ensure that any subclusters uncovered in this manner represented real cellular heterogeneity rather than artificial differences due to over-clustering, we examined the number of genes expressed differentially between them (see Materials and methods and S14 Fig). As with cluster #9, cluster #8 could also be divided into 2 subclusters, which segregated on the main UMAP. Therefore, we manually divided it (renamed cluster #8 and cluster #15) (S10L–S10Q Fig). All other subclusters were deemed to be products of over-clustering artefacts, as few genes were expressed differentially between them (S14 Fig). Clusters #11, #12, #14, #15, and #16 contained fewer than 3 cells belonging to either of the original the datasets [57,58], rendering integration, and therefore subclustering analysis, impossible.

**Validating marker gene expression in vivo.** Next, we identified marker genes enriched in each cluster that would enable us to annotate all the clusters exhaustively (see Materials and methods and Fig 5B). We validated the expression of 31 marker genes in vivo using available enhancer trap lines, antibodies, and by in situ HCR (Figs 5C–5H, S10, and S15–S17). Since whole cell morphologies were not always visible when transcripts were visualised by HCR, we validated marker gene expression in specific glial types by visualising transcript expression by HCR with glial type Gal4 lines driving GFP (S3 Fig). In this way, we annotated all 17 clusters with 13 unique glial cell identities—fenestrated glia (lamina-specific perineurial), pseudocartridge glia (lamina-specific subperineurial), chalice glia (lamina-specific perineurial), distal satellite glia (lamina-specific cortex), proximal satellite glia (lamina-specific cortex), epithelial glia (lamina-specific astrocyte), marginal glia (lamina-specific ensheathing), chiasm glia (tract ensheathing), and medulla, lobula and lobula plate perineurial, subperineurial, cortex, ensheathing, and astrocyte glia—with some glial identities mapping to multiple clusters (Figs 5C–5H and S15–S17). Hereon, non-lamina glial classes are referred to as general perineurial, general subperineurial, general cortex, general ensheathing, and general astrocyte glia.

We confirmed that clusters #9, #16, and #14 corresponded to epithelial, marginal, and proximal satellite glia, respectively, consistent with our previous comparisons to the glial cell type-specific transcriptomes (Figs 5B–5H, 6B, S16J, S17G, S17H, and S17K). Carpet glia did not appear to be represented as a unique cluster, indicating that either they are transcriptionally indistinguishable from another type of surface glia or that they were not sampled in these datasets. The latter is more likely since only 2 carpet glia are present in each optic lobe and they can be uniquely labelled with their own driver lines [70], suggesting that they are a distinct cell type. As well, inner and outer chiasm glia could not be resolved into distinct clusters though unique driver lines do distinguish between them [24]. Instead, both mapped to cluster #10. Given that these are also rare cells, it is possible that cluster #10 contains a heterogeneous population, but that insufficient cells were sampled to resolve them in the present dataset. Thus, our annotations revealed that morphological categories associated with the lamina neuropil across all glial classes were transcriptionally distinct.

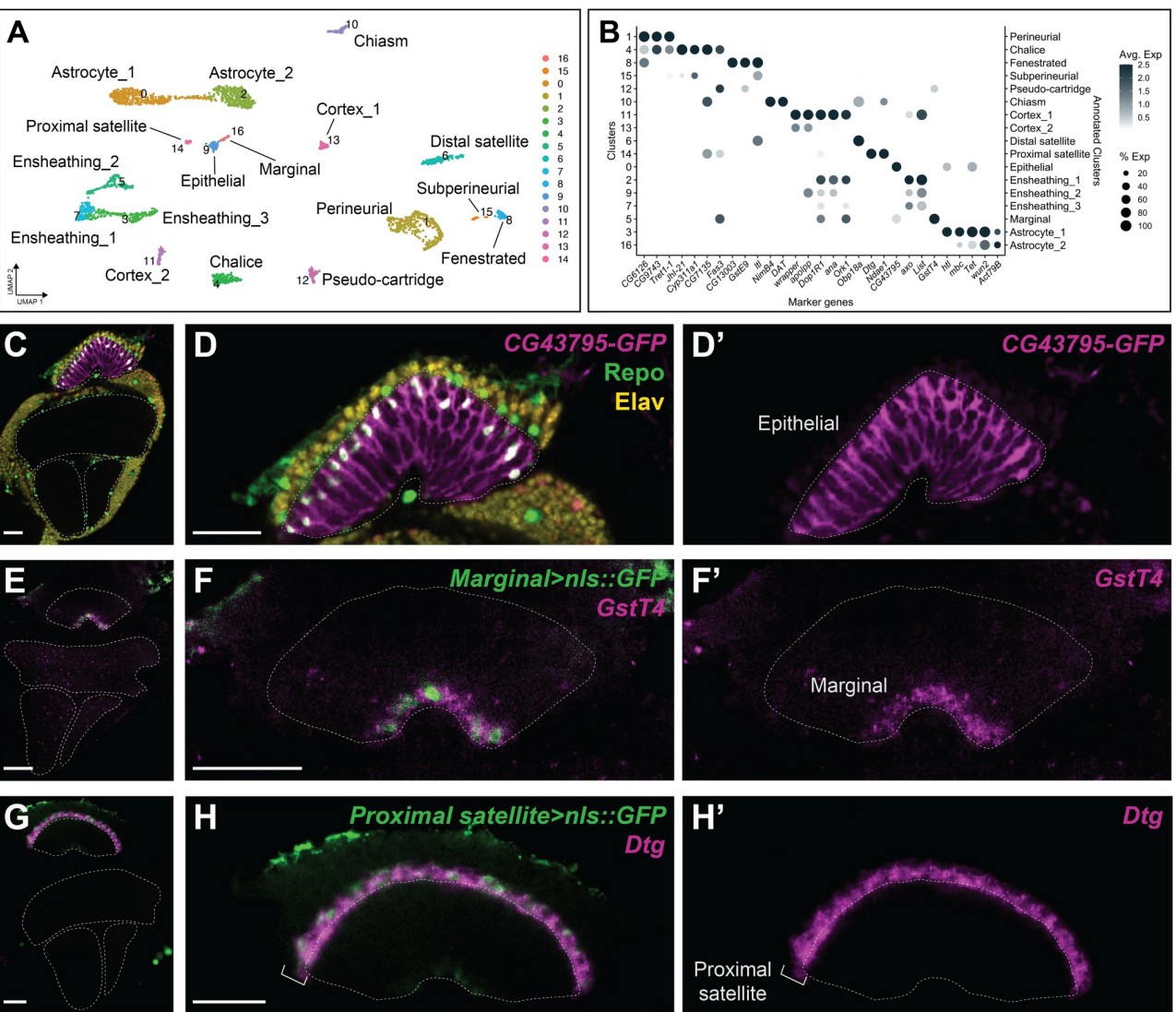

**Fig 5. Annotation of young adult optic lobe glial clusters. (A)** UMAP of the 17 clusters obtained for the young adult optic lobe dataset, labelled with both the cluster number (left) and our annotation of the specific glial class based on marker gene expression validated in vivo *(*right). **(B)** Expression plot of all marker genes selected for annotation and validated in vivo. The size of the dot represents the percentage of cells with expression in each cluster, while the colour of the dot represents the level of average expression in the cluster. **(C–H)** Three examples of marker gene validation for the whole optic lobe, focussed on the lamina: **(C, D)** *CG43795* gene trap drove GFP expression (magenta) in epithelial glia specifically. Repo in green and Elav in yellow. **(E, F)** *GstT4* and **(G, H)** *Dtg* (both in magenta) expression were detected by in situ HCR in marginal and proximal satellite glia, respectively. GFP was driven by the indicated glial-Gal4 in green. See S10 and S15–S17 Figs for additional in vivo marker gene validation. Dashed lines outline the neuropils. Single focal planes in (C–H). Scale bars are 20 μm in (C, E, G) and 5 μm in (D, F, H). The data underlying (A, B) can be found in S3 and S7 Data files. HCR, hybridization chain reaction; UMAP, uniform manifold approximation and projection.

## General optic lobe glia do not subcluster by neuropil location

While astrocytes could be categorised into 8 distinct morphologies distributed across and within neuropils (including epithelial glia of the lamina) (Fig 2A and 2Q–2W), we only identified 3 transcriptional clusters of putative astrocytes. We annotated cluster #9 as the epithelial glia (astrocytes of the lamina), whereas clusters #0 and #2 co-expressed well-known astrocyte markers including *Glutamine synthetase 2* (*Gs2*), *ebony* (*e*), *nazgul* (*naz*), *alrm*, and

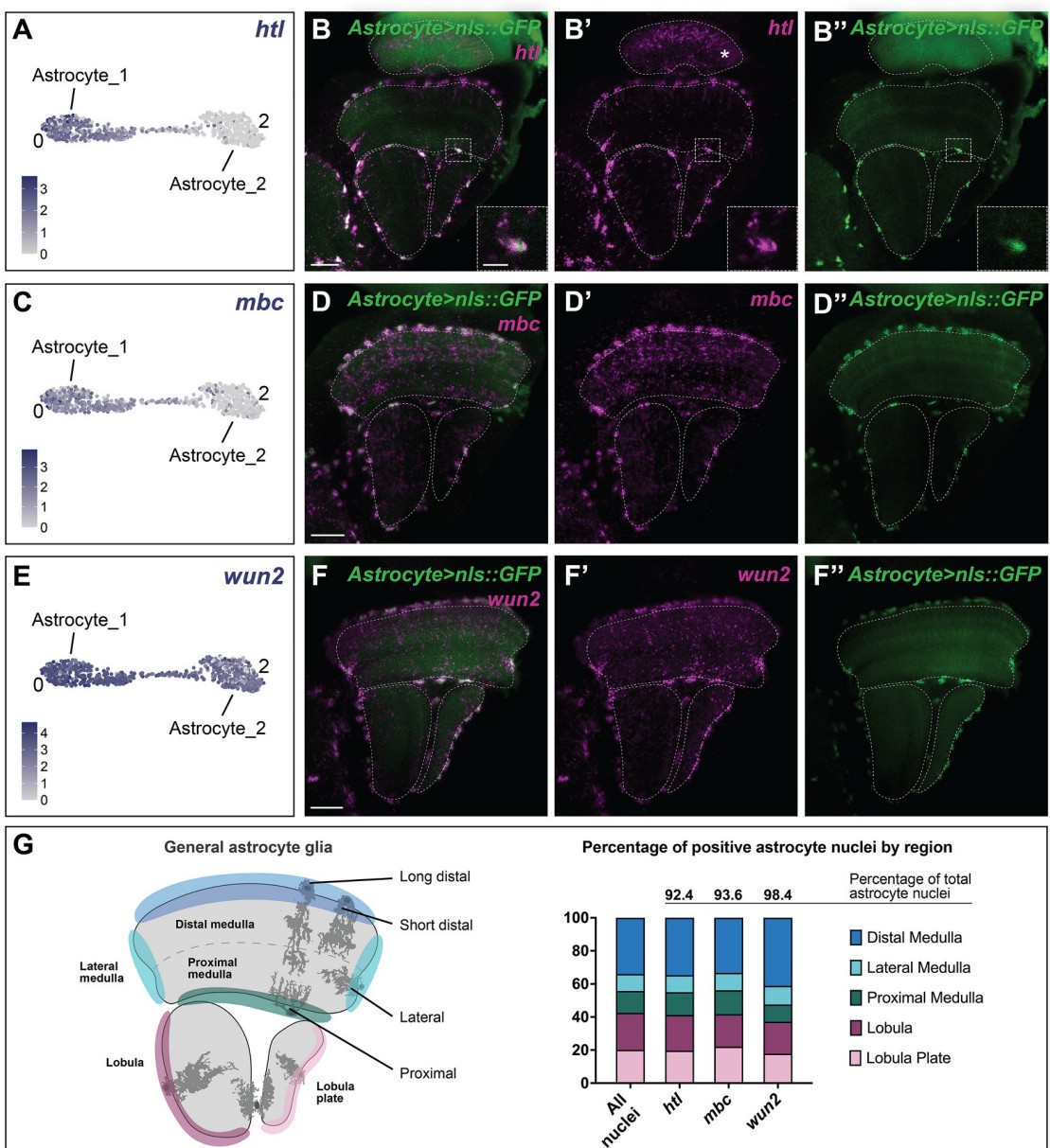

**Fig 6. Only 1 cluster within the astrocyte cluster multiplet corresponded to homeostatic astrocytes detected in vivo. (A)** *htl* expression levels plotted on a UMAP showing only cluster #0 (Astrocyte_1) and #2 (Astrocyte_2). Each dot represents a single cell, and the colour represents the level of expression as indicated. **(B)** *Astrocyte(R86E01)>nls::GFP* adult optic lobe in which all astrocyte nuclei were labelled with GFP and where *htl* expression (magenta) was detected by in situ HCR in most GFP positive nuclei. **(C)** UMAP of clusters #0 and #2 showing *mbc* expression. **(D)** *Astrocyte(R86E01)>nls::GFP* adult optic lobe where *mbc* expression (magenta) was detected by in situ HCR in most GFP (green) positive nuclei. **(E)** UMAP of cluster #0 and #2 showing *wun2* expression. **(F)** Maximum projection of *Astrocyte(R86E01)>nls::GFP* adult optic lobe where *wun2* expression (magenta) was detected by in situ HCR in most GFP (green) positive astrocyte nuclei of the medulla, lobula, and lobula plate. Dashed lines outline the neuropils and scale bars are 20 μm in (B, D, F). **(G).** Quantification of *htl*, *mbc*, and *wun2* positive (medulla, lobula, and lobula plate nuclei) astrocyte nuclei (i.e., GFP positive nuclei) from (B, D, F). More than 90% of astrocyte nuclei were positive for the 3 genes. Different regions of the medulla, lobula, and lobula plate were defined, as these captured some differences in astrocyte morphotypes. The percentage of positive nuclei within each region was quantified and no significant difference (Chi-square test; *htl* p = 0.90, *mbc* p = 0.89, *wun2* p = 0.33) was observed between the distribution of positive nuclei and the distribution of all nuclei in all 5 regions. The data underlying this figure can be found in S5 and S7 Data files and on GitHub at https://github.com/VilFernandesLab/2022_DrosophilaGlialAtlas. HCR, hybridization chain reaction; UMAP, uniform manifold approximation and projection.

*Eaat1* (and did not express *Eaat2*, a known ensheathing glia marker; S18 Fig). We wondered whether clusters #0 and #2, which were connected to each other on the UMAP visualisation, corresponded to different morphological subtypes of astrocytes or to astrocytes of different neuropils or neuropil regions. However, most differentially expressed marker genes showed enriched expression in cluster #0 compared to cluster #2; e.g., *heartless (htl)*, which encodes an Fibroblast Growth Factor receptor and *myoblast city (mbc)* were expressed in cluster #0 only. We could not find unique marker genes for cluster #2 (Fig 6A and 6C). Therefore, we examined the distribution of astrocytes across and within neuropils (excluding the lamina) using a pan-astrocyte-specific Gal4 to drive GFP expression (*R86E01>GFP*). Since *htl* and *mbc* were expressed only in cluster #0 and not cluster #2, we then compared the distribution of GFP-positive astrocytes by region (as indicated in Fig 6G) to the distribution of *htl* expressing astrocytes. Surprisingly, although *htl* and *mbc* were expressed only in one of the 2 astrocyte clusters, we detected their expression in 92.4% and 93.6% of all astrocytes, respectively, and there was no bias in their regional distribution (Fig 6A–6D and 6G). The same was true for *wunen 2 (wun 2), a* marker expressed across both clusters #0 and #2 (Fig 6E, 6F, and 6G). Therefore, both clusters mapped indistinguishably to general astrocytes, with no apparent correspondence to neuropil or morphology. Similar to clusters #0 and #2, when we validated marker gene expression in vivo, we were also unable to distinguish between clusters #3, #5, and #7, which all appeared to correspond to general ensheathing glia (i.e., ensheathing glia of the medulla, lobula, and lobula plate; S16G–S16J Fig). Likewise, clusters #11 and #13 both mapped to general cortex glia (i.e., cortex glia in the optic lobe excluding the lamina; S15B–S15D Fig). Thus, transcriptional heterogeneity of the general optic lobe glia is not regionally defined.

## Transcriptional diversity in the general optic lobe glia reflects cellular state and not cellular identity

What then is the distinction between these clusters? Across model systems, glia appear to be more susceptible to stress than neurons during tissue dissociation [57,58,60,71]. Indeed, compared with neuronal clusters, glial clusters within the Özel and Kurmangaliyev optic lobe datasets were reported to be enriched for cells with low total gene counts per cell, enriched in mitochondrial transcripts, features of low-quality transcriptomes [57,58,60]. While standard cutoffs (see Materials and methods) for total and mitochondrial gene counts were used to filter out low-quality transcriptomes, we speculated that some of the glial clusters that mapped to the same cell type (hereafter referred to as "cluster multiplets") likely represented the same cell type split by cell state (e.g., based on transcriptome quality or cellular stress). Therefore, we examined the total number of genes, the total number of reads, and the proportion of mitochondrial genes relative to the total number of genes for each cluster within a multiplet (S19A–S19C Fig). Although ensheathing glial clusters (#3, #5, and #7) and cortex glial clusters (#11 and #13) appeared to separate by transcriptome quality, we found no clear indication that general astrocyte clusters (#0 and #2) differed in this way (S19A–S19C Fig). Therefore, we performed Gene Ontology (GO) enrichment analysis on the genes differentially expressed within the general astrocyte clusters with a 1.2-fold cutoff. Interestingly, this revealed that a wide range of GO terms (from signal transduction and morphogenesis to growth and taxis) were enriched for cluster #0, whereas GO terms associated with mitochondrial regulation, autophagy, and metabolic processes were enriched for cluster #2 (S19D Fig). These data suggested that cells of cluster #2 (astrocyte_2) are unlikely to be a distinct cell type from those of cluster #0 (astrocyte_1), but instead may be cells in distinct metabolic states. Altogether, these data suggest that general astrocyte, ensheathing, and cortex

clusters may be segregating based on transcriptome quality and/or cellular state, possibly driven by tissue dissociation. Furthermore, our data indicate that clusters #0 (astrocyte_1), #3 (ensheathing_1), and #11 (cortex_1) likely correspond to a more homeostatic state of general astrocyte, ensheathing, and cortex glia, respectively. We hypothesise that the generation of cluster multiplets occurred in the adult optic lobes due to the requirement to dissect these tissues prior to dissociation, resulting in greater tissue stress compared to our whole embryo dissociations.

Since our data indicated that cluster #0 (astrocyte_1) most likely corresponded to homeostatic astrocytes of the medulla, lobula, and lobula plate, we subclustered it to further probe for any potential heterogeneity (see Materials and methods and Fig 7A). Although we obtained 3 subclusters in this way, they differed only subtly from each other. We could identify only 1 marker gene (*Actin 79B; Act79B*) that was differentially expressed by at least 4-fold among the 3 subclusters (Figs 7B and S14). Specifically, subclusters #0 and #2 showed enriched expression of *Act79B* relative to subcluster #1 (Fig 7B). Although 96% of the cells in cluster #0 (astrocyte_1; homeostatic general astrocytes) expressed *Act79B* according to the scRNA-seq data, in vivo it was expressed very sparsely, in only 10.9% of medulla, lobula, and lobula plate astrocytes (Fig 7C and 7D). These cells were found more frequently in the proximal medulla, lobula, and lobula plate (Fig 7C and 7D). *Act79B* encodes what is thought to be a muscle-specific isoform of Actin [72,73] and therefore is unlikely to be a cell-fate determinant; we speculate instead that *Act79B* may be expressed by cells in a particular state of cytoskeletal remodelling (e.g., during process extension or growth). The apparent mismatch between the proportion of *Act79B*-expressing general astrocytes in scRNA-seq and in vivo may suggest that tissue dissociation pushes cells towards a state where *Act79B* is up-regulated, but that this state is relatively rare under homeostatic conditions (i.e., *Act79B* may label a transient cell state under homeostatic conditions) (Fig 7C and 7D). In sum, the unique and stereotyped morphologies of astrocytes of the medulla, lobula, and lobula plate cannot be assigned to unique transcriptional signatures within the present depth of sequencing limits and therefore are unlikely to represent distinct subclasses of astrocytes. Furthermore, any hidden heterogeneity that we have been able to uncover within general astrocytes appears to correspond to a cell state rather than cell type or identity.

## Immune response-related genes are enriched in lamina glia

Lamina glia are the main source of transcriptional diversity in the optic lobes, with more unique cell types associated with the lamina than all other neuropils combined (Fig 8A). Indeed, in addition to its own unique counterparts for the main glial classes, the lamina contains multiple perineurial (fenestrated and chalice), subperineurial (pseudocartridge and carpet), and cortex (distal and proximal satellite) glial subtypes. To investigate lamina glia specialisation further, we performed GO enrichment analysis for biological processes on pooled lamina glia and pooled general optic lobe glia (summarised in Fig 8B and Materials and methods). These comparisons revealed that compared to general glia, lamina glia are enriched for GO terms associated with immune responses and cell junctions (Fig 8B). Indeed, among the markers for lamina glia we validated in vivo in intact optic lobes were *Tsf1*, *GILT1*, and *JhI-21*, which are known to be involved in the antibacterial and/or antifungal immune response (S17D, S17M, and S17N Fig) [74–77]. By contrast, general glia were enriched for terms associated with metabolite transport, intercellular signalling, and migration relative to lamina glia (Fig 8B). Thus, lamina glia appear to be primed to perform immune-related functions relative to their general glial counterparts.

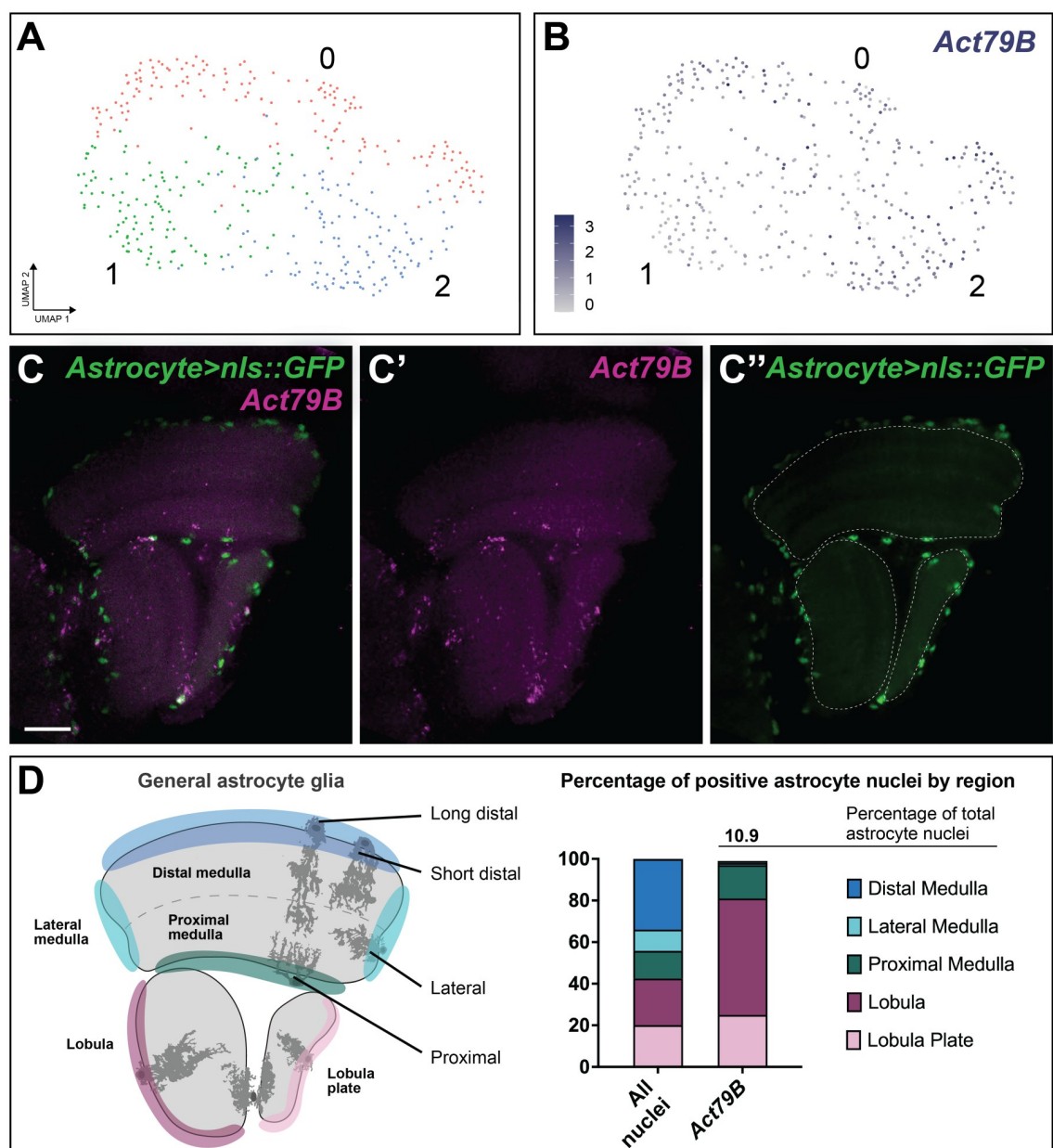

**Fig 7. Analysis of clusters obtained from subclustering cluster #0 (Astrocyte_1; homeostatic astrocytes). (A)** UMAP of the subclusters obtained from the subclustering of Astrocyte_1 (cluster #0). **(B)** *Act79B* expression levels plotted on the same UMAP. **(C)** Maximum projection of *astrocyte(R86E01)>nls::GFP* adult optic lobe where *Act79B* expression (magenta) was detected by in situ HCR in a sparse subset of GFP (green) positive astrocyte nuclei of the medulla, lobula, and lobula plate. Dashed lines outline the neuropils and scale bar is 20 μm. **(D)** Quantification of *Act79B* positive astrocyte nuclei (i.e., GFP positive nuclei) from (C) approximately 11% of astrocyte nuclei were positive for *Act79B* in vivo. The percentage of positive nuclei within each indicated region was quantified, showing *Act79B was* preferentially expressed in astrocytes of the lateral and proximal medulla, lobula, and lobula plate (Chi-squared test, *p* < 0.0001). The data underlying this figure can be found in S5 and S7 Data files and on GitHub at https://github.com/VilFernandesLab/2022_DrosophilaGlialAtlas. HCR, hybridization chain reaction; UMAP, uniform manifold approximation and projection.

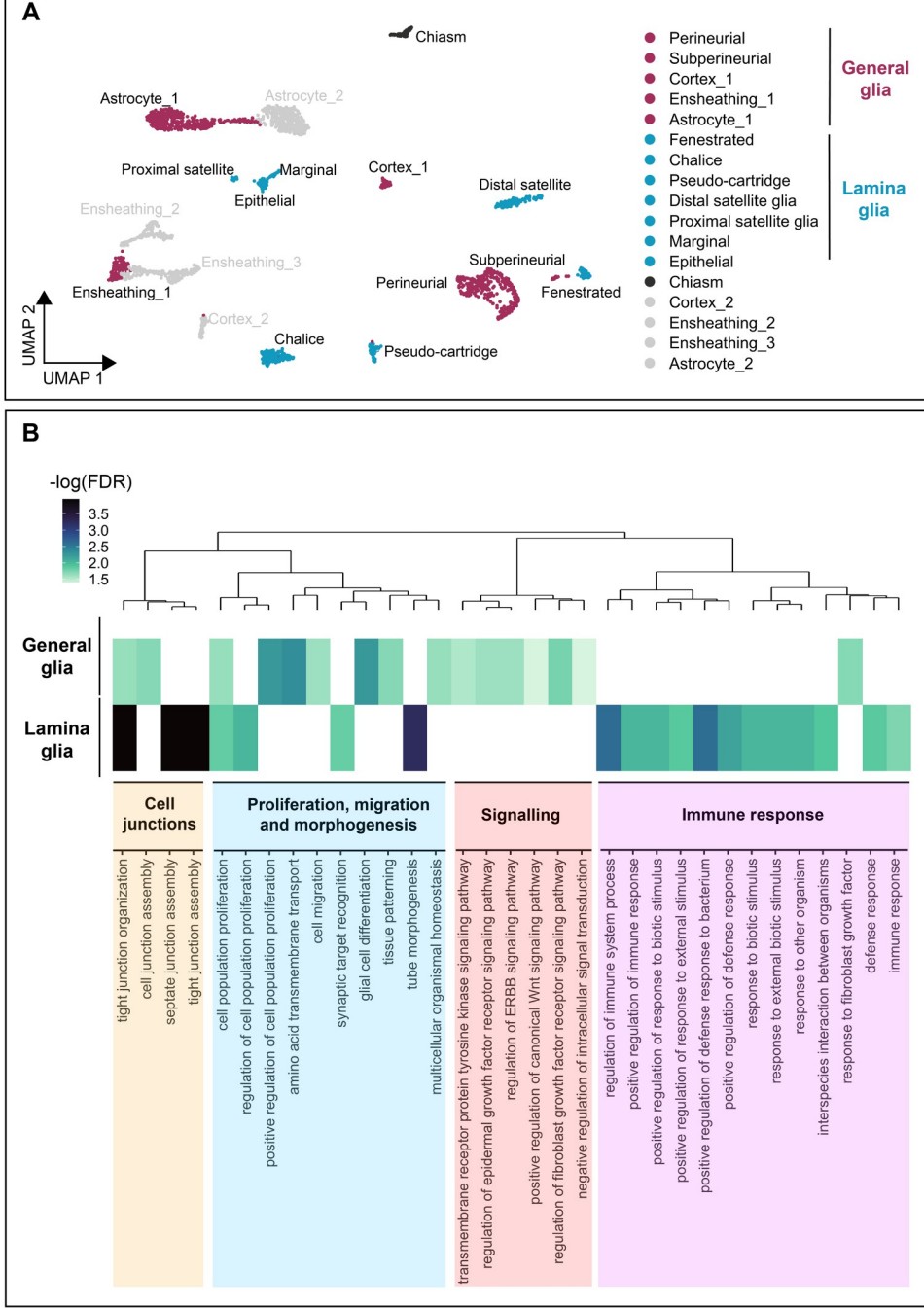

**Fig 8. Lamina glia are specialised to perform immune-related functions. (A)** UMAP of the adult optic lobe glia colour-coded for lamina (maroon) and "general" (medulla, lobula, and lobula plate; blue) glial clusters that were included in the GO analysis in (B). Note that the chiasm glia cluster (dark grey) and non-homeostatic clusters belonging to multiplets (light grey) were excluded from the analysis in (B). **(B)** GO enrichment analysis for biological processes of the lamina glia compared to other (general) optic lobe glia shown as a heatmap. We used a dendrogram to cluster GO terms into groups that were defined by a super-term. The scale indicates the negative log of the FDR. The data underlying this figure can be found in S7 Data and on GitHub at https://github.com/VilFernandesLab/2022_DrosophilaGlialAtlas. FDR, false discovery rate; GO, Gene Ontology; UMAP, uniform manifold approximation and projection.

## Motor and sensory-associated glia are transcriptionally similar

Having shown that, apart from the lamina, optic lobe glia within each class are transcriptionally similar irrespective of morphology (e.g., all astrocyte morphotypes are transcriptionally similar, with the exception of lamina astrocytes), we sought to evaluate how different glia are between the young adult optic lobe and embryo, where developmental stage, circuits, and afferent inputs are vastly different. To address this question, we integrated the embryonic dataset with the optic lobe dataset, representing motor and sensory-associated glia, respectively. Note that our embryonic dataset, though generated from whole brains, did not contain any optic lobe glia as they are not born until late larval and pupal development [54]. Strikingly, we found that the embryonic clusters and their corresponding adult general glia clusters (i.e., not the lamina clusters) converged on the UMAP visualisation (Fig 9A–9C). Indeed, several of our validated marker genes were found in both embryonic glia and their counterparts in the adult optic lobe (summarised in Fig 9D). Interestingly, there was no overlap with lamina glia clusters, except for the embryonic channel-associated perineurial glia and the chalice glia, suggesting that they may be functionally analogous (Figs 9B and 2C).

Altogether, our analysis uncovered a surprising transcriptional similarity between glia from different circuit types, brain regions, and time points in development. This suggests that, within each general class (astrocyte, cortex, ensheathing, etc.), glia likely fulfil similar functions from circuit to circuit and across neuropils.

## Discussion

We generated, validated, and annotated a single-cell transcriptional atlas of *Drosophila* glia spanning embryo to adult. In validating our dataset, we identified many new marker genes for known glial classes and subclasses and identified a transcriptionally distinct subclass of perineurial glia in the embryonic VNC, the channel-associated perineurial glia. We hope that this atlas will serve as a community resource and facilitate functional studies to unveil how distinct glial populations coordinate the development and function of neural circuits.

### Glial morphological diversity and detectable transcriptional diversity were not correlated

While we must be cautious in interpreting our results given the depth of sequencing constraints associated with scRNA-seq approaches, our data, nevertheless, point to a mismatch between glial morphological and transcriptional diversity. Specifically, glia exhibited more diversity at the morphological level than was detectable at the transcriptional level.

Although we detected striking morphological diversity in embryonic VNC subperineurial glia, this diversity could not be easily divided into discrete categories, as these glia clearly exhibited a continuum of morphologies from one extreme to the other (surface-only to channel-only) (Fig 1H–1L). Similarly, embryonic ensheathing glia also exhibited a range of morphologies from neuropil-only association to both neuropil and axon tract association to varying degrees (Fig 1N and 1O). Nonetheless, we detected only 1 transcriptional cluster corresponding to all subperineurial glia and 1 transcriptional cluster for all ensheathing glia. Even when glial morphologies could be divided into discreet categories such as for the newly hatched larval VNC and the adult optic lobe astrocytes (2 and 8 discreet morphological categories, respectively; Figs 1A, 1P–1S, 2A, 2Q–2W, and S4), detected transcriptional diversity did not match with morphological diversity. While astrocytes of the lamina were transcriptionally distinct, we identified only 1 transcriptional cluster of true, homeostatic astrocytes that corresponded to all remaining astrocyte morphologies at both developmental stages and brain

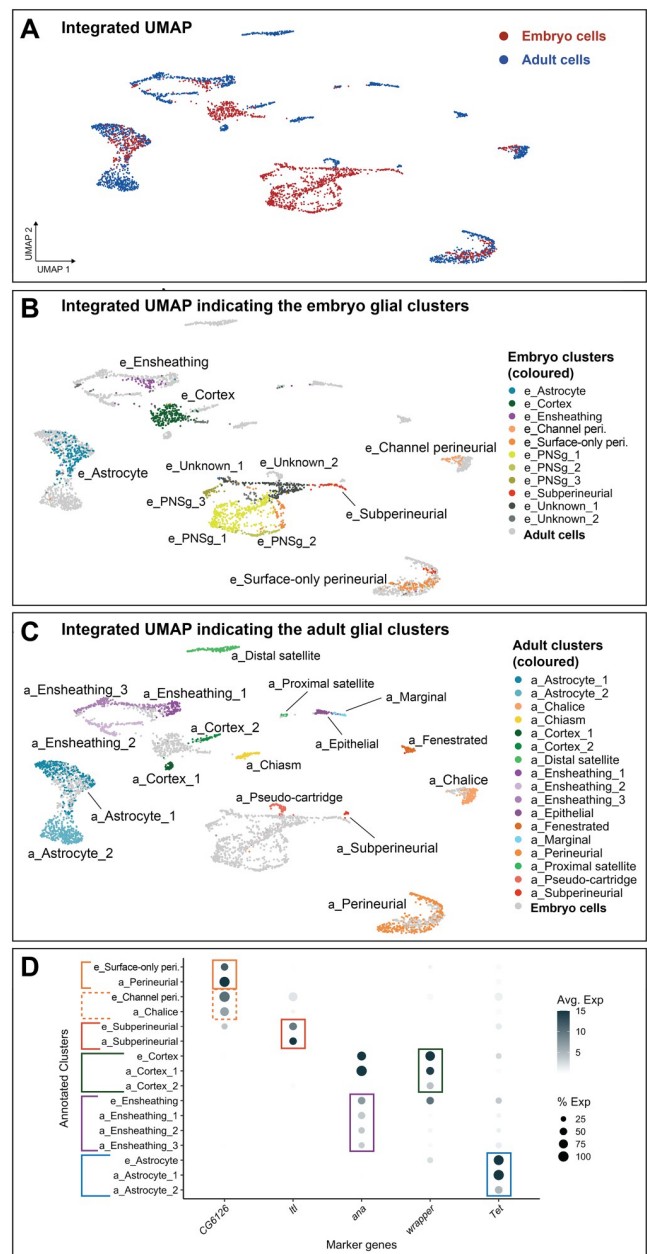

**Fig 9. Integration of the embryonic stage 17 CNS and young adult optic lobe glial clusters. (A)** UMAP of the integrated dataset, highlighting the embryonic (red) and young adult optic lobe (blue) glial clusters. **(B, C)** UMAP of the integrated dataset, highlighting embryonic (B, 'e_') or the young adult optic lobe (C, 'a_') glial clusters in colours by glial class. Grey marks cells from the young adult optic lobe clusters in (B) and from the embryonic clusters in (C). **(D)** Expression plot of genes with expression in the same glial class of the embryo and adult. The size of the dot represents the percentage of cells with expression in each cluster, while the colour of the dot represents the level of average expression in the cluster. The 5 genes indicated were validated in vivo for both time points by independent strategies: *CG6126* for perineurial (embryo only-surface perineurial, S11B, S11I, and S11J Fig; adult perineurial, S16F Fig) and chalice (S16F Fig) and channel-associated perineurial (Figs 4E, S11I, and S11J), *ltl* for subperineurial (embryo, S11E, S11F, S11I, and S11K Fig; adult, S16C–S16E Fig), *ana* for ensheathing (embryo, S13A; adult, S16 Fig), *wrapper* for cortex (embryo, Fig 4F; adult, S15B Fig), and *Tet* for astrocyte (embryo, S12B, S12D, and S12E Fig; adult, S15A Fig). The data underlying this figure can be found in S6 and S7 Data files and on GitHub at https://github.com/VilFernandesLab/2022_DrosophilaGlialAtlas. CNS, central nervous system; UMAP, uniform manifold approximation and projection.

regions. Indeed, when integrated, embryonic and adult glia co-clustered by class (except for lamina glia), despite the difference in developmental stage and origin, highlighting a striking convergence in their transcriptional profiles. This result was unexpected as the embryonic dataset was enriched for glia within a sensorimotor circuit [49–53], and the adult dataset was specific to the visual system. Thus, within the present limits of detection, our data suggest that glial classes are not transcriptionally specialised to the circuit types they associate with.

While at first glance, it may appear that some glial morphologies of the adult optic lobes were associated with unique transcriptional signatures (e.g., lamina astrocytes), the glial morphological categories with unique transcriptional signatures were all associated with the lamina. This suggests that the observed transcriptional specialisation is due to lamina association rather than specialised morphology. Interestingly, the glial transcriptional diversity we defined matches remarkably well with glial diversity described previously by glial-specific drivers from the Janelia GAL4 collection [25,78]. These driver lines also did not distinguish between cortex and ensheathing morphotypes of medulla, lobula, and lobula plate neuropils [25]. It will be interesting to explore if lineage relationships alone account for the transcriptional diversity detected in glia. Glial lineage relationships are characterised with detailed cellular resolution for the embryonic VNC where cell numbers are low, but are still incomplete for the optic lobes [24,45].

We hypothesise that glial morphology may be a plastic cellular feature that is set by the local environment and thus may represent a cellular state rather than an underlying transcriptionally defined identity. In the future, one could leverage genetic manipulations that alter neuropil structure to assess how equivalent glia might alter their morphology and transcriptional profiles in distinct environments. For example, one could assess glial morphologies and transcriptional profiles while (i) altering neuronal diversity and composition by manipulating the temporal series in medulla neuroblasts [79–82], or by blocking lamina neuron differentiation [83,84]; or (ii) by modifying neuropil stratifications with genetic perturbations that disrupt neuronal targeting [85].

## Specialisation of the *Drosophila* lamina glia

Transcriptomic studies in mammals have uncovered regional differences in astrocyte transcriptional profiles (e.g., between and within the cortex and hippocampus) [10,11]. In our optic lobe data, we could only detect within-class transcriptional specialisation for glia associated with the lamina, whereas we could not detect any within-class transcriptional differences between (or within) the other optic lobe regions. This is particularly surprising since others have speculated that circuit complexity (stemming from neuronal diversity) may drive glial specialisation [6,86]; however, the lamina is the simplest of all neuropils, while the medulla is the most complex [55]. Thus, at least for the optic lobes, region-specific neuronal diversity does not appear to be a predictor of glial transcriptional diversity, within the present limits of detection.

Instead, our data revealed an up-regulation of GO terms related to immune functions in lamina glia when compared to other optic lobe glia (Fig 8B). The lamina is the first neuropil to receive input from photoreceptor sensory neurons [55,87] and, as such, its surface glia are densely perforated to allow photoreceptor axons to enter the lamina [22]. Thus, the lamina may be subject to more pathogenic insults compared to other optic lobe neuropils. This finding opens up the intriguing hypothesis that glia that interface with the environment may be specialised to manage environmental insults (e.g., heat, cold, lesions). Recently, a specialised population of cutaneous Schwann cells was identified which sense noxious stimuli and initiate pain sensation [88]. Furthermore, Schwann cells can migrate away from peripheral nerves to

aid in cutaneous wound healing following injury, which is accompanied by an up-regulation of genes related to the innate immune response [89]. Defining the extent to which other glial cell populations at environmental boundaries adopt immune-related roles will broaden our understanding of the functional properties of glia not only for nervous system function, but also organismal health.

### Limitations of this study

It is possible that all within-class glial morphological categories are transcriptionally distinct, but that the genes that confer this diversity are lowly expressed and therefore were missed due to depth of sequencing limitations. It is also possible that functional specialisation may occur locally in micro-domains of the same cell. For example, astrocytes are intimately associated with neuronal synapses. At the synapse, astrocytes regulate synapse strength and turnover of local neurotransmitters. As a single astrocyte supports many distinct types of synapses simultaneously, each requiring different molecular machinery [86], astrocytes must express a wide variety of channels and receptors (resulting in transcriptional homogeneity). Functional diversity could therefore be conferred by local synaptic activity. In other words, a single astrocyte may have functionally distinct microdomains dictated by the synapses it supports. Indeed, recent studies show that synapse activity regulates positioning of neurotransmitter receptors in astrocyte processes [90–92]. Functional studies, including in vivo Calcium, Glutamate, or GABA imaging will help to resolve glial functional specialisation both within and across cells.

A major anatomical difference between *Drosophila* and mammals is the size of glial domains relative to the size of the brain. Although only 10% to 15% of the *Drosophila* brain is made up of glia, astrocytes tile the entire CNS. For example, 6 individual astrocytes are sufficient to tile a single hemisegment of the larval VNC [26]. In contrast, thousands of astrocytes are required to tile an analogous section of the mouse spinal cord [93]. As mammalian astrocytes proportionally tile a much smaller region of the nervous system, it is conceivable that a single mammalian astrocyte can interact with one circuit type. In contrast, *Drosophila* astrocytes frequently span multiple circuits, which may require more functional flexibility. Thus, it is possible that further exploration of astrocyte diversity in vertebrates will identify unique transcriptional programs that specify morphologically distinct astrocytes with regionalized identities and functions. In sum, our work highlights the necessity for a systematic multimodal approach to characterising glial diversity in other systems [94,95].

## Materials and methods

### *Drosophila* stocks and maintenance

*Drosophila melanogaster* strains and crosses were reared on standard cornmeal medium and raised at 25˚C unless stated otherwise. We used the following strains in this study (see S3 File for more details):

hs-FLPG5;; 10×UAS(FRT.stop)myr::smGdP-HA, 10×UAS(FRT.stop)myr::smGdP-V5, 10×UAS(FRT.stop)myr::smGdP-Flag (hsMCFO; BDSC 64085), pBPhsFlp2::PEST in attP3;; HA_V5_FLAG_OLLAS (BDSC: 64086), CG6126-Gal4 (BDSC 67505), CG5080-Gal4 (BDSC 83275), CG10702/CG17343-Gal4 (BDSC 76234), Ntan1$^{Mz97}$-Gal4 (BDSC 9488), PRL-1-Gal4 (BDSC 65566), R54H02-Gal4 (BDSC 45784), Eaat2-Gal4 (BDSC 78932), alrm-Gal4 (BDSC 67031 and 67032), Rdl-Gal4 (BDSC 65421 and 66509), moody-Gal4 (BDSC 90883), R25H07-Gal4 (BDSC 49145), Tet-Gal4 (BDSC 19427), pum-Gal4 (BDSC 63368), w[*] TI{RFP [3xP3.cUa] = TI}Tre1[attP] (BDSC 84582), ana-Gal4 (BDSC 86394), CG9657-Gal4 (BDSC 78971), CG9449-Gal4 (BDSC 91274), pippin-Gal4 (BDSC 86401), ltl-Gal4 (BDSC 76144), PRL-Gal4 (BDSC 81151), hs-flp;lexAop-myr::tdTomato; repo-lexA, R85G01-Gal4 (BDSC:

40436), *R54C07-Gal4* (BDSC: 50472), *R10C12-Gal4* (BDSC: 47841), *R47G01-Gal4* (BDSC: 45768), *R50A12-Gal4* (BDSC: 47618), *R46H12-Gal4* (BDSC: 50285), *R53H12-Gal4* (BDSC: 50456), *R56F03-Gal4* (BDSC: 39157), *R35E04-Gal4* (BDSC: 48127), *R86E01-Gal4* (BDSC: 45914), *55B03-Gal4* (BDSC: 39101), *Repo-Gal4* (BDSC: 7415), Mi{MIC}CG43795-GFP [MI03737] (BDSC: 41395), 10XUAS-mCD8::GFP (BDSC 32184 and 32186), 10XUAS-IVS-myr::GFP (BDSC 32197) and UAS-nls::GFP (BDSC 4775 and 4776).

## Embryo collections

For timed collections of embryonic and larval stages, crosses were reared in collection bottles fitted with 3% agarose apple caps coated with yeast paste. Embryos were then collected for 1.5-hour (h) windows and reared at 25˚C until the desired developmental stage.

## Multicolour flip-out clonal analyses (see S3 File for list of genotypes)

*Embryonic/larval ventral nerve cord*: To generate MCFO clones, we crossed virgin females of the hsMCFO line to males of Gal4 lines that were known to be expressed [26,47,96,97] or found to be enriched in a given transcriptional cluster (see S3 File for full genotype list). Embryos were collected for 1.5 h, aged for 6 h (embryonic stage 11/12, prior to gliogenesis [65]), and then heatshocked at 37˚C for 15 min to induce FLP expression. Embryos were then transferred to 4˚C for 5 min to halt FLP expression and transferred to 25˚C until hatching. At 25˚C, hatching occurs at 21 h after egg laying [98].

*Adult optic lobe*: For MCFO experiments of adult brains, we utilised Janelia Gal4 driver lines with specific glial expression, as previously described [25] (S3 Fig). We crossed males of the glial-Gal4 lines with virgin females of a 3-tag (HA, V5, and FLAG) MCFO line, except the cortex and marginal Gal4s which we crossed with a 4-tag (HA, V5, FLAG, and OLLAS) MCFO (S3 File). We raised progeny at 18˚C and induced FLP expression in adult flies (0 to 5 days old) by heat-shocking at 37˚C. Length of heat shocks varied depending on the Gal4 driver (S3 File). After heat-shocking, flies were placed back at 18˚C for at least 1 night, before being dissected and stained as detailed in our immunohistochemistry protocol.

## Immunohistochemistry

*Larval VNC*: For VNC studies, larvae were dissected at 0 h after larval hatching. We dissected larval brains in sterile-filtered, ice-cold 1× PBS. Brains were then mounted on 12 mm #1 thickness poly-d-lysine coated round coverslips (Neuvitro Corporation, GG-12-pdl) and fixed in fresh 4% paraformaldehyde (Electron Microscopy Sciences, 15710) in 1× PBS with 0.3% Triton detergent (0.3% PBST). We then washed the coverslips in 0.3% PBST to remove fixative and blocked overnight at 4˚C in 0.3% PBST supplemented with 1% BSA (Fisher, BP1600-100), 1% normal donkey serum, and 1% normal goat serum (Jackson ImmunoResearch Laboratories 017-000-121 and 005-000-121). Brains were then incubated in primary antibody overnight at 4˚C, washed overnight at 4˚C with 0.3% PBST, and then incubated in secondary antibodies overnight at 4˚C. We then removed the secondary antibodies, transferred the coverslips to 0.3% PBST overnight, and mounted in DPX. To mount in DPX, brains were dehydrated with an ethanol series: 30%, 50%, 70%, and 90%, each for 5 min, then twice in 100% ethanol for 10 min each (Decon Labs, 2716GEA). Samples were then transferred to glass-bottomed depression slides with xylenes (Fisher Chemical, X5-1) for 2 × 10 min. Finally, samples were mounted onto slides containing DPX mountant (Millipore Sigma, 06552) and cured for 1 to 2 days before imaging.

*Adult optic lobe*: For immunocytochemistry experiments of the adult optic lobe, we dissected whole brains in 1× phosphate buffered saline (PBS), fixed in 4% paraformaldehyde for

30 min, then washed with 0.5% PBTx (1× PBS with 0.5% TritonX). We next incubated the samples with primary antibodies diluted in block (5% normal horse serum), for 2 nights. Samples were washed with 0.5% PBTx, incubated for a further 2 nights with secondary antibodies diluted in block, washed again, and mounted in SlowFade (Life Technologies).

We used the following primary antibodies: Rabbit anti-Pros (1:1,000) [99], Rat anti-HA (1:100; Millipore Sigma, 11867423001), Chicken anti-V5 (1:1,000; Bethyl Laboratories, A190-118A), Mouse anti-Repo (1:50, DSHB 8D12), Mouse anti-Cherry (1:500; Clontech, 632543), Rabbit anti-Gat (1:4,000; a gift from M. Freeman) [23], Rat anti-FLAG (1:400, Novus NBP1-06712), Rabbit anti-HA-tag (1:400, Cell Signalling Technology, C29F4), Rat anti-Elav (1:20, DSHB 7E8A10), Mouse anti-V5-Tag:DyLight-550 (1:400, Bio-Rad), Chicken anti-GFP (1:400, EMD Millipore), Rabbit anti-GFP (1:400, Thermo Fisher A6455), DyLight 405 conjugated Goat anti-Horseradish peroxidase (HRP) (1:50, Jackson ImmunoResearch, 123-475-021), Rabbit anti-dsRed (1:500, Takara, 632496), and Mouse anti-FasIII (1:20, DSHB 7G10).

We used the following secondary antibodies at 1:400: Alexa Fluor Rhodamine Red-X Donkey anti-Mouse (Jackson ImmunoResearch, 715-295-151), Alexa Fluor 488 Donkey anti-Rabbit (Jackson ImmunoResearch, 711-545-152), DyLight 405 Donkey-anti Rabbit 405 (Jackson ImmunoResearch, 711-475-152), Alexa Fluor 488 Donkey anti-Chicken (Jackson ImmunoResearch, 703-545-155), Alexa Fluor Rhodamine Red-X Donkey anti-Rat (Jackson ImmunoResearch, 712-295-153), Alexa Fluor 647 Donkey anti-Mouse (Jackson ImmunoResearch, 715-605-151), and Alexa Fluor 647 donkey anti-Rat (Jackson Immunolabs, 712-605-153).

## Embryo dissociation for single-cell RNA sequencing (scRNA-seq)

We prepared cell dissociates from embryos at 17 to 18.5 h after egg laying (stage 17). We washed embryos in deionized water before surface sterilising them in 30% bleach for 2 min. We then homogenised them in Chan-Gehring (C + G) medium by 6 to 8 strokes of a loose-fitting dounce. We filtered the cell suspension through a 40 μm Nitex mesh and pelleted cells in a clinical centrifuge at 4°C (setting 5, IEC). We washed the cell pellet twice by pouring off the supernatant and gently resuspending the pellet in fresh C + G, pelleting between each rinse as above. We determined the cell-survival proportion for each dissociate using the BioRad TC-20 trypan-blue assay. Samples that met a threshold of 80% viability were submitted for sequencing at a concentration of 1,000 cells per microliter.

## Single-cell RNA sequencing of stage 17 embryos

The University of Oregon Genomics and Cell Characterization core facility (https://gc3f.uoregon.edu/) prepared embryonic cell samples for scRNA-seq. We ran dissociated cells on a 10X Chromium platform using 10X NextGem v3.1 chemistry targeting 10,000 cells per sample. Following cDNA library preparation, we amplified the library with 15 cycles of PCR before sequencing on 2 separate Illumina Hi-seq lanes, providing 2 technical replicates. Following examination for batch effects between technical replicates (S5 Fig), we merged the datasets using the CellRanger Aggregate function prior to quality control and downstream analysis. Reads were aligned to the *Drosophila* genome (BDGP6.22) and protein coding reads were counted.

## Initial quality control and cell clustering of glia from embryonic scRNA-seq

We analysed the resulting sequencing data with the 10X CellRanger pipeline, version 3.1.0 [100], R version 3.6.3 and Seurat [101] version 3.1.2 using standard quality control, normalisation, and analysis steps. We filtered cells by the percentage of mitochondrial genes expressed

(relative to the total number of genes expressed), indicating a high stress state. Only cells expressing <10% mitochondrial reads were retained for analysis. Additionally, cells containing reads for <50 and >3,000 unique genes were filtered out of downstream analysis. For each gene, expression levels were normalised by total expression, multiplied by a scale factor (10,000) and log-transformed, and the top 3,000 variable genes were identified for downstream principal component analysis (PCA). Clustering was performed using 50 PCs [*FindClusters* resolution 1.0], resulting in a Seurat object containing 52,881 cells. To isolate neuronal and *repo+* glial cell clusters from all somatic tissues, we visualised the expression of *elav*, *repo*, *Gat*, and *alrm* in UMAP space. We then selected embryonic clusters 18, 2, 0, 4, 24, 26, 22, 16, 13, 14, 11, and 27 to be subset for further analysis. The resulting subset of cells were re-clustered using the previous parameters, resulting in the stage 17 embryonic neurons and *repo+* glia dataset (19,600 cells). To isolate embryonic *repo+* glia from embryonic neurons, we again visualised the expression of *elav*, *repo*, *Gat*, and *alrm* in UMAP space. From this object, clusters 23, 26, 34, 38, 17, 41, 27, 32, and 19 were identified as glial/non-neuronal. All the scripts described above are available at https://github.com/AustinSeroka/2022_stage17_glia. The resulting dataset (3,221 cells) was exported (.Rds) for subsequent analyses.

Midline glia are known to express *sim* and *wrapper*, but do not express *elav* or *repo* [35]. Therefore, to identify midline glia from the full dataset, we visualised *sim*, *wrapper*, *elav*, and *repo* expression in UMAP space. We identified a small group of cells meeting these expression criteria, which were clustered together and belonged to a larger cluster (cluster 19). We selected these cells as putative midline glia and defined them as a new cluster within the whole embryonic dataset. We pooled midline glia together with the stage 17 embryonic neurons and *repo+* glia dataset described above and performed hierarchical clustering analysis [*hclust*] on this combined dataset, using average cluster gene expression to produce a Euclidean dissimilarity matrix [*dist*]. We used [*FindAllMarkers*] (default parameters, except only.pos = TRUE) to obtain the most highly expressed genes for putative midline glia relative to all other embryonic cells (see S1 Data).

## Preliminary integration of published optic lobe glial single-cell RNA sequencing datasets

Two independent studies published scRNA-seq datasets of the optic lobes throughout pupal development with a focus on neuronal development and therefore did not analyse glial clusters [57,58]. We took advantage of the glial cells within these datasets here. Özel and colleagues performed scRNA-seq on optic lobes at specific stages in triplicate, profiling a total of 275,000 cells [57], whereas Kurmangaliyev and colleagues performed multiplexed scRNA-seq of many developing brain samples in parallel and profiled 51,000 cells in total [58]. The 2 datasets (hereafter referred to as the Özel and Kurmangaliyev datasets, respectively) thus varied dramatically in the number of cells they each profiled. Here, we chose to focus specifically on optic lobe glia from developmental stages that most closely match the young adult, i.e., young adult dataset from [57] and 96h APF dataset from [58] We used R (version 4.1.0) and Seurat (version 4.0.3) [101] to analyse the optic lobe scRNA-seq datasets from NCBI GEO accession GSE142787NCBI GEO accession GSE156455 [57,58], and [] enclose the specific Seurat functions used.

We converted the Kurmangaliyev 10X Genomics data to an RDS file [*CreateSeuratObject*] and subsetted [*subset*] the cells belonging to the 96h APF time point (within metadata, under "time"). Glial clusters were subsetted from both the 96h APF-Kurmangaliyev dataset (within metadata, under "class") and adult-Özel dataset based on the original annotation; i.e., *repo* expression. Since the 2 datasets represent glia from the same structure and from very close

developmental stages, we used Seurat's Integration pipeline to find corresponding cells and batch correct the 2 datasets (Fig 3). We normalised each glial dataset [*NormalizeData*] and selected the 2,000 most variable features [*FindVariableFeatures*]. We defined Anchors [*FindIntegrationAnchors*, dims = 1:30] and integrated the datasets [*IntegrateData*, dims = 1:30]. We then scaled the integrated dataset [*ScaleData*] and ran PCA [*RunPCA*, npcs = 30]. We checked elbow plots [*ElbowPlot*] to determine the number of dimensions to use in [*FindNeighbors*, dims = 1:16], followed by [*FindClusters*, resolution = 0.5]. We obtained 20 clusters and used a UMAP dimensionality reduction [*RunUMAP*, reduction = pca, dims = 1:20] to plot the cell clusters and visualise their distribution as a 2D representation [*DimPlot*] (Figs 3, S7C, and S7D).

### Further clean-up of young adult optic lobe and embryonic glial datasets

**Neuronal and hemocyte clean-up.** While analysing the adult optic lobe dataset, we noticed that most glial clusters contained cells that formed streams that extended towards the centre of the UMAP and that cells in these streams expressed high levels of *Rdl*, *Frq1*, *Nckx30C*, *elav*, and *fne*, whereas cells belonging to the main body of clusters did not express these genes (S8A Fig). *elav* and *fne* are well-documented pan-neuronal marker genes. Indeed, we found that *Rdl*, *Frq1*, and *Nckx30C* genes were also expressed pan-neuronally (S8B Fig). Thus, we hypothesised that they likely represented contamination from neurons due to the close association of glia and neurons. We also observed *Rdl*, *Frq1*, *Nckx30C*, *elav*, and *fne* co-expressing cells in the embryonic glial dataset (mainly from cluster 4) and pan-neuronally (S8C and S8D Fig). We validated that *Frq1* and *Nckx30C* are expressed neurons and not glia by HCR in the adult optic lobes (S9A–S9C Fig), and similarly through MCFO clonal analyses in the embryo found that *Rdl* was exclusively expressed in neurons (S9D Fig). We then examined the distribution of cells expressing different levels of *Rdl*, *Frq1*, and *Nckx30C* in individual glial clusters and all neuronal clusters averaged [*RidgePlot*] (S9E and S9F Fig) and used these distributions to choose a cutoff to eliminate *Rdl*, *Frq1*, and *Nckx30C* expressing cells from the glial clusters [*subset*, *Rdl/ Frq1/ Nckx30C* ≤ 1] (5,101 cells in total; S9G and S9H Fig). To remove contaminated cells in the embryo, we used the same thresholds (S9J–S9N Fig) as the adult dataset (346 cells in total).

We also eliminated putative hemocytes from both the young adult optic lobe (3 cells) and embryonic (10 cells) glial datasets by removing cells that expressed *Hml*, a hemocyte-specific marker [*subset*, *Hml* ≤ 0].

### Re-clustering the embryonic glial dataset

Following the clean-up described above for the embryonic glial dataset, we normalized the data [*NormalizeData*], selected the 2,000 most variable features [*FindVariableFeatures*], scaled the data [*ScaleData*], and ran PCA [*RunPCA*, npcs = 30]. We examined elbow plots [*ElbowPlot*] to determine the number of dimensions to use in [*FindNeighbors*, dims = 1:16], followed by [*FindClusters*, resolution = 0.5]. We obtained 11 clusters and plotted them as a 2D representation ([*RunUMAP*, reduction = pca, dims = 1:20]; [*DimPlot*]; Fig 3; see also S9L–S9N Fig). We used [*FindAllMarkers*] (default parameters, except only.pos = TRUE) to obtain the most highly expressed genes for the embryonic glia (see S2 Data).

### Re-integrating the young adult optic lobe dataset and eliminating clusters originating from a single dataset

Following the clean-up described above for the young adult optic lobe glial dataset, we obtained a list of the cells remaining using [*WhichCells*] (all identities, by default *ident*

argument) and isolated these cells from the original 96h APF-Kurmangaliyev and young adult-Özel datasets (using [*subset*] and lists of cells used as *cells* argument). We then integrated these as described previously, [*FindNeighbors*, dims = 1:19] and [*FindClusters*, resolution = 0.5] (S9I Fig). We obtained 19 clusters. Since others have reported that glial clustering is sensitive to batch effects [57,58,60], we eliminated clusters to which the Kurmangaliyev dataset contributed fewer than 1% of the total number of cells in the cluster (S9I Fig; a total of 850 cells were eliminated) and reintegrated the remaining cells as described above (Fig 3). Following this, the young adult optic lobe glial dataset consisted of 15 clusters (Fig 3).

## Comparing cell type-specific bulk RNA sequencing to scRNA-seq

Cell type-specific transcriptomes, obtained by Tandem-affinity purification of intact nuclei sequencing, were published for the marginal glia, epithelial glia, and the proximal satellite glia [68] (NCBI GEO accession #GSE116969). We simulated the FACS-sorted glial type-specific transcriptomes as single-cell transcriptomes to enable us to compare these deep bulk RNA sequencing datasets with the shallower scRNA-seq dataset, as described in [63]. Briefly, a random number of reads were assigned to each of 900 simulated cells, from a normal distribution with the same mean and standard deviation as the cells in the final Adult-96hAPF dataset. The probability of expression of each gene was obtained by dividing the number of transcripts per million (TPM) of each gene by the total, in each cell-type, and this probability was used to allocate the number of reads of each simulated cell. The simulated matrix of expression was transformed into a RDS object, normalised [*NormalizeData*] and average expression of each gene was calculated [*AverageExpression*]. We then examined the Pearson correlations between the average expression of the simulated single-cell and the average expression of the scRNA-seq clusters to determine the best match.

## Subclustering individual glial clusters

To analyse individual young adult optic lobe glial clusters, we used [*WhichCells*] to obtain the list of cells belonging to a specific cluster and subsetted those cells from the original Kurmangaliyev and Özel datasets [*subset*]. We then integrated these as described above. The arguments *dims*, *k.score* and *k.filter* in [*FindIntegrationAnchors*], and *dims* and *k.weight* in [*IntegrateData*], were assigned a value of 1 unit lower than the smallest number of cells in either the Kurmangaliyev or Özel subsetted cells. Next, we used [*FindNeighbors*] and [*FindClusters*] (see Fig S14 for specific dimensions and resolutions used).

## Adding subclusters to main young adult optic lobe glial dataset

S14 Fig summarises the number of subclusters obtained following subclustering of individual glial clusters. To account for over-clustering artefacts, we examined the number of genes that were differentially expressed between subclusters. Only the subclusters of clusters 8 and 9 expressed greater than 20 genes differentially (4-fold change or higher), whereas all other subclusters expressed fewer than 15 genes differentially. Therefore, we proceeded to manually divide clusters 8 and 9 accordingly (S10F–S10Q Fig). We obtained lists of cells belonging to each of these subclusters using [*WhichCells*] and used these lists to define new cell clusters in the full dataset, thus generating 17 clusters in total (Fig 5A).

## Finding markers for clusters in the embryonic and young adult optic lobe glial datasets

We identified marker genes expressed by each glial cluster using [*FindMarkers*]. To find genes with highly specific expression, we focused on the top 30 differentially expressed transcripts. We then examined their expression within and across clusters [*FeaturePlot*] to choose transcripts that were expressed by most cells in a given cluster (see Figs 4B and 5B for a summary of the marker genes validated in vivo). We used [*FindAllMarkers*] (default parameters, except only.pos = TRUE) to obtain the most highly expressed genes for the young adult optic lobe glia (see S3 Data).

## In situ hybridisation chain reaction (HCR) probe design

To assess the expression of marker genes in vivo, we designed HCR probes against chosen marker genes (Fig 5B). We designed 6–21 antisense probe pairs against each target gene, tiled along the annotated transcripts but excluding regions of strong sequence similarity to other transcripts (see S4 Data), with the corresponding initiator sequences for amplifiers B3 and B5 [102]. We purchased HCR probes as DNA oligos from Thermo Fisher (at 100 μm in water and frozen).

## In situ hybridisation chain reaction (HCR)

We dissected optic lobe–central brain complexes from adult flies (female and male) in 1× PBS. We then fixed them in 4% formaldehyde for 35 min at room temperature. We then rinsed (3×) and washed (3× 30 min) them in PBSTx (1× PBS with 0.5% Triton X-100, Fisher BioReagents). Next, we transferred the optic lobe–central brain complexes to 1.5 mL Eppendorf tubes and followed the Multiplexed HCR RNA-FISH protocol (Detection and Amplification stages) for whole-mount fruit fly embryos, from Molecular Instruments (molecularinstruments.com), also described in [102] with the following modifications: we used 2 pmol of stock probe set and 12 pmol of each hairpin. Before proceeding with the 30 min washes with 5× sodium chloride sodium citrate (SSC) with 0.1% Tween 20 (SSCT), we incubated samples with in 5× SSCT (with DAPI) for 2 h at room temperature. All HCR buffers and hairpins were purchased from Molecular Instruments. Samples were stored at 4˚C and mounted in SlowFade Gold Antifade Mountant (Thermo Fisher) within 3 days of completing the protocol.

## Microscopy and image processing

*Larval VNC*: We used a Zeiss LSM700 point-scanning or an Intelligent Imaging Innovations (3i) spinning disc confocal microscope with a 63× objective to image all samples with a step size of 0.2 to 0.38 μm and the following laser lines: 405, 488, 555, and 647. Images were acquired using Zen Black or Slidebook, respectively. Samples were imaged to encompass the entire dorsal-ventral extent of the VNC.

*Young adult optic lobe*: We used a Zeiss LSM800 or LSM880 confocal microscope with 20× or 40× objectives to image whole optic lobes with a maximum step size of 1 μm. Individual astrocytes for morphological quantifications were imaged on a Zeiss LSM980 confocal microscope with AiryScan 2. Briefly, we acquired z-stacks (step size 3 to 4 μm) of whole optic lobes containing isolated MCFO-labelled astrocytes to determine astrocyte cell body position (using anti-Repo) along with information on the neuropil layers occupied by individual astrocytes (using anti-Brp). We then acquired z-stacks (optimal step size) of individual astrocytes at high resolution.

## Quantifications and statistical analyses

*Embryonic astrocyte morphological quantifications*: To quantify the volume of type 1 versus type 2 astrocytes, we used Bitplane Imaris (version 10.0.0) to render 3D constructions of individual astrocyte MCFO clones (genotype: *hsMCFO + alrm-Gal4*). We used the *[Surfaces]* function with default parameters to measure total cell volume. Primary branches were quantified manually by segmenting astrocytes, 5 microns at a time, using the Imaris [*Ortho Slicer*] function.

*Adult optic lobe astrocyte volumetric quantifications*: We used Bitplane Imaris (version 9.9.0) to render 3D reconstructions of individual astrocytes. We used the [*Surfaces*] function with default parameters to measure total cell volume. We then aligned cells using the [*Frame*] function and cut the object (the cell) by neuropil layers (binning 2 layers together), and measured the volume for each was section of the cell. We quantified 8 cells of each astrocyte morphological category. Primary branches were quantified manually by segmenting astrocytes as detailed above. We used Graphpad Prism 9 to analyse the data for statistical significance as indicated in the main text.

*Larval CNS*: In Bitplane Imaris (version 9.6.1), a 3D projection was created for each individual VNC. For quantification of Repo+ cells in the VNC versus brain lobes, we used the Imaris "Spots" function to automatically reconstruct and quantify the number of Repo+ nuclei (xy diameter set to 3 μm, z spread set to 5 μm, manual thresholding). We quantified Repo+ cells in 7 independent VNC samples and 7 independent brain lobe samples at hatching (Genotype: *hsMCFO + fne-Gal4*). VNC enrichment was then determined via a one-way ANOVA in GraphPad Prism 9. For morphometric analyses, subtypes and morphotypes were quantified manually, blinded to genotype, and assessed statistically in Graphpad Prism 9 by Chi-squared statistical tests.

*Adult optic lobe astrocyte cluster validation*: We used the following approach to quantify the proportion of total astrocytes expressing a particular marker gene and their positions within the optic lobe. In Fiji (ImageJ2 version 3.2.0), we chose 35 to 50 optical slices [*Make Substack*] from each Z-stack. We performed a standard background subtraction [*Subtract Background*] (default parameters). We then created an average intensity projection and measured the mean fluorescence intensity of the HCR probe channel to obtain a Normalisation value (see below). We then used Icy (version 2; [icy.bioimageanalysis.org](http://icy.bioimageanalysis.org)) for further analysis. In Icy, we defined neuropil regions (as in Figs [6] and [7] for astrocytes), we then used the *Spot Detector* plugin (spot scale of 13 or 25 pixels in size and 40 to 100 sensitivity) to identify glial nuclei marked by nuclear GFP (*R86E01-Gal4>nls::GFP*) within these neuropil regions. We manually removed nuclei from the central brain that were detected by this method to restrict our analysis to the medulla, lobula, and lobula plate only. We then quantified the mean fluorescence intensity of mRNAs detected by HCR in the defined ROIs. We scaled these measures to the normalisation value (defined above) and used a threshold of 2.25, above which cells were considered positive for expression of the specific transcript examined. We settled on this threshold empirically by analysing several samples for markers that these cells were known to express or not express. We then calculated the proportion of nuclei positive for a given transcript in each optic lobe (excluding the lamina) as well as the proportion of positive nuclei within different neuropil regions. We used GraphPad Prism 9 to analyse these data with Chi-squared statistical tests.

## Gene Ontology enrichment analyses

All GO enrichment analyses were carried out in R studio (version 1.4.1717), using R (version 4.1.1). We identified differentially expressed genes in our cluster/s of interest using [*FindMarkers*] from Seurat (version 4.1.1) [101]. For comparisons of one single cluster to another,

markers were selected with a log2FC $\geq$ 0.25 (fold change of 1.2). When comparing larger groups of clusters (i.e., the pooled lamina glia clusters to the pooled general glia clusters), a more stringent threshold of 1 log2FC (fold change of 2) was used to select differentially expressed genes. GO enrichment analysis for biological processes (BP) were carried out using [enrichGO] from clusterProfiler (version 4.2.2) [101]. Within this function, the adjusted $p$-values associated with the GO terms were calculated, using the Benjamini–Hochberg (BH) adjustment method for multiple comparisons. Only terms with adjusted $p$-value $< 0.05$ were considered. We next used [*simplify*] from clusterProfiler [103], with a p-adjust threshold of 0.7, which acts to reduce the redundancy in the enriched GO terms. The top 20 GO terms for each group were selected based on their significance (adjusted $p$-value) and plotted as a heatmap using ggplot2 (version 3.3.5) [104]. A dendrogram of the enriched GO terms was built using the Ward D agglomerative method with [*hclust; dist*], based on their pairwise similarity calculated with the Wang method using [*pairwise_termsim*] from the enrichplot package (version 1.14.2). This dendrogram was used to arrange and group the GO terms, enabling us to manually define super terms for the enrichment. For several poorly annotated GO terms, we examined the annotated genes and manually renamed the GO terms (see R script available on our GitHub page for full details).

## Supporting information

**S1 Fig. Channel-associated perineurial glia characterisation. (A)** Lateral view of the VNC at 0 h after larval hatching showing channel-associated perineurial glia (cyan and magenta) on the ventral and dorsal surfaces, each sending a single projection with ventral channel-associated perineurial glia sending longer processes than their dorsal counterparts. **(B–C')** Lateral view (B) and cross-sectional view (C) of the VNC at 0 h after larval hatching showing a single ventral channel-associated perineurial clone (cyan), and all other glia in red. Note in (B) the red outer glial membranes along the channels that belong to enveloping channel-associated subperineurial glia. Individual channel-associated perineurial glial cells have their surface domains along one side or the other of the midline (C). All scale bars represent 10 μm. **(D)** Box and whisker plots of the area at the main surface of the VNC occupied by surface-only perineurial glia, channel-associated perineurial glia and surface-only subperineurial glia. $N \geq 6$ clones from $N \geq 6$ brains. Line in the middle represents the median; box limits represent the 25th and 75th percentiles; whiskers indicate minimum and maximum. Mann–Whitney U-test $p$-values indicated (ns, non-significant; * $p < 0.05$; ** $p < 0.005$). The data underlying this panel can be found in S5 Data.
(TIF)

**S2 Fig. Quantifications of newly hatched VNC astrocyte morphologies. (A)** Schematic of the cross-section of the embryonic VNC showing the different astrocyte positions and morphologies. Position types are dorsal, lateral, and ventral based on nucleus position, separated by dashed lines (left). Type 1 morphologies are more arborised and type 2 are less arborised. Morphologies described originate from sparse *alrm>MCFO* clones. **(B)** Proportions of type 1 and type 2 morphologies for astrocyte clones at 0 h after larval hatching with nuclei in dorsal, lateral, or ventral positions, as defined in (A). Type 1 astrocytes are more prevalent laterally, while type 2 astrocytes are more prevalent in dorsal and ventral positions. $N = 117$ clones from $N = 36$ brains. **(C, D)** Box and whisker plots of total cell volumes (C) and number of primary branches (D) for astrocyte clones in the dorsal, lateral, and ventral positions. Line in the middle represents the median; box limits represent the 25th and 75th percentiles; whiskers indicate minimum and maximum. The lateral astrocytes, mainly type 1, show higher volumes and more primary branches compared to dorsal and ventral astrocytes, mainly type 2. $N \geq 6$ clones

from $N \geq 6$ brains per neuropil position. Mann–Whitney U-test $p$-values indicated (ns, non-significant; ** $p < 0.01$; *** $p < 0.001$). The data underlying (B–D) can be found in S5 Data.
(TIF)

**S3 Fig. Expression patterns of the glial Gal4 drivers used to evaluate glial morphology in the *Drosophila* adult optic lobe. (A–M)** GFP expression driven by the indicated glial Gal4 driver (described previously in [25]) for the **(A, B)** optic lobe perineurial glia, (C, D) optic lobe subperineurial glia, (E) optic lobe astrocyte glia, (F) optic lobe ensheathing glia (not including chiasm glia; also drives expression in a subset of neurons), (G) chiasm glia (and marginal glia), (H) medulla, lobulla, and lobula plate cortex glia and proximal satellite glia, (I) lamina astrocytes (epithelial glia), (J) lamina ensheathing glia (marginal glia), (K) lamina-specific cortex glia (proximal satellite glia), (L) lamina subperineurial glia (pseudo-cartridge glia), and (M) lamina perineurial glia (fenestrated glia). (N) A table outlining the glial Gal4 lines and the glial subtypes that they drive expression within. Where relevant, cyan marks CD8::GFP driven by the glia-Gal4, magenta marks Repo, yellow labels Elav and HRP labels the neuropils in white. Panels B and D are maximum projections showing the surface of the optic lobe. Dashed lines outline the neuropils. All scale bars represent 20 μm.
(TIF)

**S4 Fig. Morphological quantifications of adult optic lobe astrocytes. (A)** Schematic of the cross-section of the adult optic lobe and its four neuropils: lamina, medulla, lobula, and lobula plate, with the different astrocyte morphologies indicated. Morphologies described originate from sparse *Astrocyte(R86E01)>MCFO* clones. Dashed lines indicate the neuropil layers. **(B, C)** Box and whisker plots of total cell volumes (B) and number of primary branches (C) for each astrocyte morphology. $N = 8$ clones. **(D, E)** Box and whisker plots of cell volumes within each neuropil layer pair for each astrocyte morphology of the medulla (D) and lobula complex (E). $N = 8$ clones. Line in the middle represents the median; box limits represent the 25th and 75th percentiles; whiskers indicate minimum and maximum. The data underlying (B–E) can be found in S5 Data.
(TIF)

**S5 Fig. Examining batch effects between technical and biological replicates for embryonic scRNA-seq. (A)** Schematic of data analysis for stage 17 whole embryo; 10X libraries were prepared from 4 biological replicates, each sequenced across 2 separate lanes. Following examination for batch effects **(B–F)** using the 8 sets of resulting CellRanger outputs, CellRanger Aggregate was used to merge the data. The resulting CellRanger outputs (Sarah_aggr_trial1_barcodes.tsv, Sarah_aggr_trial1_features.tsv, Sarah_aggr_trial1_matrix.mtx) were then read into Seurat for downstream processing (see Materials and methods). **(B)** Schematic of data analysis to check for batch effects between sequencing lanes. As this analysis predates Seurat integration functionalities, the stage 17 embryo data generated from sequencing lane 1 (biological replicates 1–4) and sequencing lane 2 (biological replicates 1–4) were combined into 2 Seurat objects, merged and clustered **(C)**. The number of cells derived from each sequencing lane were calculated for each cluster in UMAP space, and the correlation between the 2 sequencing lanes was calculated ($R^2 = 0.995$) (see "220706 glia paper supp fig 1.Rmd"). **(D)** Lane-of-origin was plotted onto the clusters in UMAP space, showing an even distribution across all clusters. Together, these analyses demonstrate the absence of batch effects between sequencing lanes. **(E)** Schematic of data analysis for examining batch effects between biological replicates. Reads for each biological replicate were combined across both sequencing lanes. These 4 objects were then merged using Seurat and clustered using standard methods. Biological-replicate-of-origin for each cell was plotted onto the clusters in UMAP space. Cells derived

from each of the 4 biological replicates contributed equally to the neuronal (illustrated by *elav* expression) and glial (illustrated by *repo* expression) clusters of the UMAP **(F)**, demonstrating the absence of batch effects across the replicates. The data underlying this figure can be found at NCBI GEO accession GSE208324 and https://github.com/AustinSeroka/2022_stage17_glia. (TIF)

**S6 Fig. Annotation of the cluster corresponding to the midline glia within the whole embryo dataset. (A–D)** Expression levels of *elav* (A), *repo* (B), *sim* (C), and *wrapper* (D) plotted on the whole embryo UMAP. Each dot represents a single cell, and the colour represents the level of expression as indicated. Zoomed-in details of cluster 19 are shown. **(E)** UMAP of the whole embryo, indicating clusters defined as midline glia (purple), neurons (blue), and *repo*+ glia (yellow), based on *elav*, *repo*, *sim*, and *wrapper* expression. **(F)** Dendrogram of hierarchical clustering average expression of all genes between midline glia (purple), neurons (blue), and *repo*+ glia (yellow) clusters. Midline glia form an outgroup to both neuron and *repo*+ glia. **(G, H)** Single focal planes of MCFO clones (magenta) of midline glia generated with *wrapper-Gal4*. HRP in cyan. Scale bar is 10 μm. The data underlying this figure can be found at NCBI GEO accession GSE208324, https://github.com/VilFernandesLab/2022_ DrosophilaGlialAtlas and S1 Data. (TIF)

**S7 Fig. Contribution of the 2 datasets, 96h APF, and 3-day-old adult, to the resulting integrated young adult optic lobe dataset. (A)** UMAP of the 19 glial clusters from 3-day-old adult optic lobes, from [57]. **(B)** UMAP of the 19 glial clusters from 96h APF optic lobes, from [58]. **(C, D)** UMAP of the integrated young adult optic lobe dataset, highlighting the 3-day-old adult clusters (C) or 96h APF clusters (D) in colour, with 96h APF (C) and 3-day-old adult (D) cells in grey. The data underlying this figure can be found at NCBI GEO accessions GSE142787 and GSE156455, and https://github.com/VilFernandesLab/2022_ DrosophilaGlialAtlas. (TIF)

**S8 Fig. Contamination of glial clusters with cells with typical neuronal profile. (A)** Expression levels of *Rdl*, *Frq1*, *Nckx30C*, *elav*, and *fne* plotted on the young adult optic lobe integrated UMAP, before neuronal clean-up. Each dot represents a single cell, and the colour represents the level of expression as indicated. Zoomed-in details of the centre of the UMAP are shown for *Nckx30C*, *elav*, and *fne*. **(B)** Expression levels of *elav*, *Rdl*, *Frq1*, and *Nckx30C* plotted on the 3-day-old adult optic lobe UMAP, from [57], including all clusters except the 19 glial clusters. All 4 genes showed expression in all clusters, illustrating the pan-neuronal nature of *Rdl*, *Frq1*, and *Nckx30C* expression. **(C)** Expression levels of *Rdl*, *Frq1*, *Nckx30C* plotted on the embryonic glial UMAP, before neuronal clean-up. The expression of these 3 genes overlapped with the expression of *elav* and *fne*, mainly in cluster #4. **(D)** Expression levels of *Rdl*, *Frq1* and *Nckx30C elav* and *fne* plotted on the UMAP of the embryonic nervous system clusters. The data underlying this figure can be found at NCBI GEO accessions GSE142787 and GSE156455, and https://github.com/VilFernandesLab/2022_DrosophilaGlialAtlas. (TIF)

**S9 Fig. Clean-up of contamination of glial clusters with cells with typical neuronal profile. (A)** *Frq1* (yellow) and *Nckx30C* (magenta) expression were detected by in situ HCR in the cortex area of the optic lobe. GFP labelled all glial cells in green. Single focal plane. Scale bar is 20 μm. **(B, C)** Single focal planes showing glial somas (green) with *Frq1* (yellow) and *Nckx30C* (magenta) expression. DAPI marks all nuclei in white. Dashed lines outline glial somas. Scale bars are 5 μm. **(D)** Single focal planes of MCFO clones (cyan) generated with *Rdl-Gal4*. Repo

in green and Prospero in magenta. Dashed lines outline the neuropil. Scale bar is 7 μm. **(E)** The distribution of *Rdl* expression levels in each glial cluster of the young adult optic lobe integrated dataset, before neuronal clean-up. **(F)** The distribution of *Rdl* expression levels in all cells of the 3-day-old adult optic lobe UMAP, from [57], except glial clusters. **(G)** UMAP of the 20 glial clusters obtained from the first integration of optic lobe datasets. **(H)** UMAP of 19 glial clusters after clean-up of potential neurons by excluding cells with normalised expression of *Rdl*, *Frq1*, and *Nckx30C* >1, and cells with *Hml* expression (>0 normalised expression) as potential hemocytes. **(I)** Cluster #3, #6, and #8 were excluded from the UMAP in (H) since less than 1% of the cells contained in them originated from the Kurmangaliyev dataset (see Materials and methods for details). **(J)** The distribution of *Rdl* expression in each glial cluster of the embryonic dataset, before neuronal clean-up. **(K)** The distribution of *Rdl* expression in all cells of the embryonic nervous system, excluding cells belonging to the 19 glial clusters. **(L)** UMAP of the 19 initial glial clusters of the embryonic nervous system, before neuronal clean-up. **(M)** Same UMAP as in (L) after exclusion of potential neurons (normalised expression >1 of *Rdl*, *Frq1*, and *Nckx30C*) and potential hemocytes (normalised expression >0 of *Hml* expression). **(N)** UMAP of the remaining cells in (M) after reanalysis and reclustering (see Materials and methods for details). The data underlying this figure can be found at NCBI GEO accessions GSE208324, GSE142787 and GSE156455, GitHubs: https://github.com/AustinSeroka/2022_stage17_glia and https://github.com/VilFernandesLab/2022_DrosophilaGlialAtlas, and in S6 and S7 Data files.
(TIF)

**S10 Fig. Subclustering of young adult optic lobe glial clusters to find hidden rare subtypes. (A)** UMAP of the 15 glial clusters of the integrated young adult dataset. **(B)** Schematic of the adult optic lobe lamina with marginal, epithelial, and proximal satellite glia represented. **(C–E)** Pearson correlation values between the bulk-RNA-seq transcriptomes of the proximal satellite, marginal and epithelial glia and the top 5 glial cluster matches from the young adult dataset. **(F)** Subclustering of cluster #9 generated 2 subclusters. **(G)** *GstT4* (a gene highly expressed marginal glia bulk RNA-seq data) showed high expression in subcluster #1. **(H)** *CG43795* (a gene highly expressed in epithelial glia bulk RNA-seq) showed high expression in subcluster #0. **(I, J)** Expression of GstT4 (I) and CG43795 (J) in the UMAP of the 15 glial clusters of the integrated young adult dataset. Both genes showed mutually exclusive expression patterns within cluster #9. **(K)** UMAP showing the original cluster #9 split into new clusters #9 and #16 (see Materials and methods). **(L)** Subclustering cluster #8 generated 2 subclusters. **(M, N)** *Gli (Gliotactin)* and *Mdr65 (Multi drug resistance 65)*, both known markers of subperineurial glia [105], were expressed in subcluster #1. **(O, P)** Expression of *Gli* and *Mdr65* on the UMAP of the 16 glial clusters of the integrated young adult dataset. Both genes showed overlapped expression in a group of cells belonging to cluster #8. **(Q)** UMAP showing the original cluster #8 split into new clusters #8 and #15 (see Materials and methods). The data underlying this figure can be found in S3, S5, and S7 Data files.
(TIF)

**S11 Fig. Validation and analysis of new markers for *Drosophila* VNC surface glia. (A)** Schematic of the cross-section of the embryonic VNC showing the different surface glial types, channel-associated and surface-only perineurial or subperineural glia. **(B–H)** Surface and cross-sectional views of MCFO clones at 0 h after larval hatching generated with the Gal4 lines indicated, belonging to marker genes with high expression in the surface glia clusters: *CG6126* (*N* = 372 clones from *N* = 11 brains), *PRL-1* (*N* = 154 clones from *N* = 17 brains), *ltl* (*N* = 19 clones from *N* = 7 brains), and *Ntan1* (*N* = 68 clones from *N* = 14 brains). All MCFO clones labelled in cyan and magenta, with Repo in green. **(I)** Quantification of the frequency of clones

recovered by glial class for the indicated driver. **(J)** Quantification of perineurial glia morpho-type frequency by driver line. Channel-associated perineurial glia were detected in 100% of *CG6126* MCFO brains (*N* = 11 brains total) and 92.9% of *CG5080* brains (*N* = 13 brains total), compared to 7.7% of *pippin* MCFO brains (*N* = 13 brains total) and 5.7% of *PRL-1* MCFO brains (*N* = 15 brains total). **(K)** Quantification of subperineurial glia morphotype frequency by driver line. Channel-associated subperineurial glia were detected in 90.9% of *moody* MCFO brains (*N* = 12 brains total), 28.6% of *ltl* MCFO brains (*N* = 7 brains total), 85.7% of *Ntan1* MCFO brains (*N* = 15 brains total), 100% of *CG10702* MCFO brains (*N* = 15 brains total), and 40% of *PRL* MCFO brains (*N* = 15 brains total). Dashed lines outline the neuropil. All scale bars are 10 μm. The data underlying (I–K) can be found in S5 Data.
(TIF)

**S12 Fig. Validation and analysis of new markers for *Drosophila* VNC astrocytes. (A–C)** MCFO clones (VNC cross sections) at 0 h after larval hatching generated with the Gal4 lines indicated, belonging to marker genes with high expression in the astrocyte cluster: *Fur1* (*N* = 27 clones from *N* = 9 brains), *Tet* (*N* = 71 clones from *N* = 10 brains), and *pum* (*N* = 25 clones from *N* = 9 brains). All MCFO clones labelled in cyan, with Repo in green and Prospero in magenta. Insets in (A–C) show Prospero and Repo in glial nuclei; only astrocyte clones were positive for Prospero. Dashed lines outline the neuropils. **(D)** Quantification of the frequency of glial type clone for each indicated driver line: *alrm* (*N* = 117 clones from *N* = 36 brains), other Ns noted above. **(E)** Quantification of astrocyte morphotype frequency by driver line. **(F)** VNC cross section showing colocalization of the astrocyte marker Gat (cyan) and a gene-trap line where 3xRFP has been swapped for the Tre1 locus (red). Dashed lines outline the neuropil. Scale bars are 10 μm. The data underlying (D, E) can be found in S5 Data.
(TIF)

**S13 Fig. Validation and analysis of new markers for *Drosophila* VNC ensheathing and cortex glia. (A–C)** MCFO clones (VNC cross sections) at 0 h after larval hatching generated with the Gal4 lines indicated, belonging to marker genes with high expression in the ensheathing or cortex clusters: *ana* (*N* = 388 clones from *N* = 11 brains), *CG9657* (*N* = 287 clones from *N* = 10 brains), and *CG9449* (*N* = 5 clones from *N* = 3 brains). Tract and neuropil ensheathing types are indicated. All MCFO clones labelled in cyan and blue, with Repo in green and Prospero in magenta. Insets in (A, B, C) show Prospero and Repo in glial nuclei, where only astrocyte clones were positive for Prospero. Dashed lines outline the neuropils. **(D)** Quantification of the frequency of glial type clone for each indicated driver line: *wrapper* (*N* = 13 brains), other Ns noted above. **(E)** Quantification of the frequency of brains with clones of both ensheathing types, tract and neuropil, or only neuropil clones, for each indicated driver line. Dashed lines outline the neuropil. Scale bars are 10 μm. The data underlying (D, E) can be found in S5 Data.
(TIF)

**S14 Fig. Summary of the number of differentially expressed genes (DEGs) obtained for different fold-change (FC) cutoffs between subclusters within each of the 15 glial clusters post clean-up of the young adult glial dataset.** The data underlying this figure can be found in S7 Data and https://github.com/VilFernandesLab/2022_DrosophilaGlialAtlas.
(TIF)

**S15 Fig. Validation of marker genes for astrocyte, cortex, and chiasm glia of the young adult optic lobe. (A)** *Astrocyte(R86E01)>nls::GFP* adult optic lobe where *Tet (Ten-Eleven Translocation (TET) family protein)* expression (magenta) was detected by in situ HCR throughout the cortex, including in many GFP positive astrocyte (green) nuclei (outlined by thicker dashed lines). See Figs 5 and 6 for additional in vivo astrocyte marker gene validation.

**(B)** *Cortex(R54H02)>nls::GFP* adult optic lobe where *wrapper* expression (magenta) was detected by in situ HCR in many GFP positive cortex glia (green) nuclei. Inset of zoomed in nuclei (scale bar is 5 μm). **(C)** *Cortex(R54H02)>nls::GFP* adult optic lobe where *apolpp (apolipophorin)* expression (magenta) was detected by in situ HCR in most GFP positive (green) nuclei. A few nuclei adjacent to the neuropil were also positive (likely ensheathing glia based on the scRNA-seq data). **(D)** *Cortex(R54H02)>nls::GFP* adult optic lobe where *Ork1 (Open rectifier K+ channel 1)* expression (magenta) was detected by in situ HCR in some GFP positive cortex glia (green) nuclei. Many nuclei adjacent to the neuropil were also positive (likely ensheathing glia based on the scRNA-seq data), as well as in marginal glia. **(E)** *Chiasm (R53H12)>GFP* adult optic lobe where *NimB4 (Nimrod B4)* expression (magenta) was detected by in situ HCR in most GFP positive chiasm glia (green) nuclei. Inset of zoomed in nuclei (scale bar is 5 μm). **(F)** *Chiasm(R53H12)>GFP* adult optic lobe where *DAT (Dopamine transporter)* expression (magenta) was detected by in situ HCR in most GFP positive chiasm glia (green) nuclei. Some nuclei in the cortex region around the medulla were positive for *DAT* (asterisk), consistent neuronal expression in the scRNA-seq data from [57]. Single focal planes in (A, E, F) and maximum projections of 11–12 focal planes (1 μm each) in (B–D). Dashed lines outline the neuropils and scale bars are 20 μm.
(TIF)

**S16 Fig. Validation of marker genes for surface and ensheathing glia of the young adult optic lobe.** **(A, B)** *Perineurial(R85G01)>GFP* adult optic lobe where **(A)** *Tret1-1 (Trehalose transporter 1–1)* and **(B)** *CG9743* expression (magenta) was detected by in situ HCR. For both genes, most GFP positive (green) nuclei of the general perineurial glia (surrounding the medulla, lobula, and lobula plate) were positive, in addition to the chalice glia (as predicted by the scRNA-seq data). *Tret1-1* was shown to be expressed in perineurial glia [106]. **(C)** Adult optic lobes expressing myrGFP driven by *ltl(larval translucida)-Gal4*, gene trap Trojan line, showing *ltl* expression in general surface glia (arrow), distal satellite (bar), and lamina surface (asterisk; fenestrated glia as predicted by the scRNA-seq data). **(D, E)** Maximum projection of *ltl*-Gal4-labelled MCFO clones showing subperineurial morphology (see Fig 2). **(F)** *Perineurial (R85G01)>GFP* adult optic lobe where *CG6126* expression (magenta) was detected by in situ HCR. Most GFP positive (green) nuclei of the general perineurial glia were positive, in addition to the chalice and fenestrated glia, as predicted by the scRNA-seq data. **(G)** *Ensheathing (R56F03)>GFP* adult optic lobe where *axo (axotactin)* expression (magenta) was detected by in situ HCR in most GFP positive (green) nuclei adjacent to the medulla, lobula, and lobula plate neuropils. Inset of zoomed-in nuclei (scale bar is 5 μm). **(H)** *Ensheathing(R56F03)>GFP* adult optic lobe where *List* expression (magenta) was detected by in situ HCR in most GFP positive (green) nuclei adjacent to the medulla, lobula, and lobula plate neuropils, and in some GFP negative nuclei in the cortex area of the same neuropils (asterisk) (general cortex glia as predicted by the scRNA-seq data). **(I)** *Ensheathing(R56F03)>GFP* adult optic lobe where *ana (anachronism)* expression (magenta) was detected by in situ HCR in most GFP positive (green) nuclei adjacent to the neuropil as well as many nuclei in the cortex area, which were probably cortex glia based on the large size of the nuclei (predicted by scRNA-seq data). **(J)** *Ensheathing(R56F03)>GFP* adult optic lobe where *Dop1R1 (Dopamine 1-like receptor 1)* expression (magenta) was detected by in situ HCR in many GFP positive (green) nuclei adjacent to the medulla, lobula, and lobula plate neuropils as well as nuclei in the cortex area of the medulla, lobula and lobula plate, marginal glia and proximal satellite glia (predicted by scRNA-seq data). All panels are single focal planes unless stated otherwise. Dashed lines outline the neuropils and scale bars are 20 μm.
(TIF)

**S17 Fig. Validation of marker genes for lamina-specific glial subtypes of the young adult optic lobe. (A, B)** *Chiasm(R53H12)>GFP* adult optic lobe, showing expression in chiasm and distal satellite glia. *Obp18a* expression (magenta) was detected by in situ HCR in most GFP positive (green) nuclei of chiasm and distal satellite glia. **(C, D)** *Perineurial(R85G01)>GFP* adult optic lobe where *JhI-21 (Juvenile hormone Inducible-21)* expression (magenta) was detected by in situ HCR, specifically in chalice glia. **(E, F)** *Perineurial(R85G01)>CD8::GFP* adult optic lobe where CG13003 expression (magenta) was detected by in situ HCR in fenestrated glia. **(G)** *Chiasm(R53H12)>GFP* adult optic lobe where *CG7135* expression (magenta) was detected by in situ HCR in chiasm, chalice, and proximal satellite glia. **(H)** Fas3 (magenta) antibody staining of *Fenestrated(R47G01)>CD8::GFP* adult optic lobe, showed expression in the lamina surface glia layer below the fenestrated glia, pseudo-cartridge glia, in the chalice, and proximal satellite glia. **(I)** *Perineurial(R85G01)>CD8::GFP* adult optic lobe where *Cyp311a1 (Cytochrome P450 311a1)* expression (magenta) was detected by in situ HCR, specifically in chalice glia. **(J)** *Fenestrated(R47G01)>nls::GFP* adult optic lobe where *GstE9 (Glutathione S transferase E9)* expression (magenta) was detected by in situ HCR in fenestrated and pseudo-cartridge glia. **(K)** *Proximal satellite(R46H12)>nls::GFP* adult optic lobe where *Ndae1 (Na+-driven anion exchanger 1)* expression (magenta) was detected by in situ HCR in proximal satellite glia, as well as in L1, L2, L3, and L4 lamina neurons, as predicted by the neuronal scRNA-seq data from [57]. **(L)** Large MCFO clones labelled by *ltl-Gal4*, marked distal satellite (bar) and fenestrated glia (arrow). HRP in white. **(M)** *Chiasm(R53H12)>CD8::GFP* adult optic lobe where *Tsf1* expression (magenta) was detected by in situ HCR in proximal and distal satellite and chiasm glia. **(N)** *Proximal satellite(R46H12)>CD8::GFP* adult optic lobe where *GILT1* expression (magenta) was detected by in situ HCR in proximal satellite glia, as well as in T1 medulla neurons, as predicted by the neuronal scRNA-seq data from [57]. All panels are single focal planes. Dashed lines outline the neuropils and scale bars are 20 μm.
(TIF)

**S18 Fig. Expression levels of known astrocytic markers and *Eaat2* ensheathing marker, in the young adult optic lobe glial clusters. (A–C)** Plots showing *Gs2*, *e*, and *naz* expression levels in young adult optic lobe glial clusters. Each dot represents a single cell, and the colour represents the level of expression as indicated. As previously described [107], *Gs2* was expressed in both astrocytes (clusters indicated in a box in A) and ensheathing (clusters indicated with asterisk in A), while *e* and *naz* are expressed exclusively in astrocytes. **(D)** *alrm*, a known astrocyte marker [24], showed expression in the astrocyte clusters (box in A). **(E)** *Eaat1*, a known astrocyte marker [26], showed expression in the astrocyte clusters (box in A). **(F)** *Eaat2*, a known ensheathing marker [26], showed expression in the ensheathing clusters (asterisk in A) and no expression in the astrocyte clusters (box in A). The data underlying this figure can be found in S7 Data.
(TIF)

**S19 Fig. Assessment of the transcriptome quality of "multiplet" clusters. (A–C)** Violin plots outlining the number of features, number of counts, and proportion of mitochondrial genes, for the astrocyte **(A)**, ensheathing **(B)**, and cortex **(C)** glia cluster multiplets. Cluster identity is colour-coded and the cluster number is indicated on the x-axes. **(A)** Astrocyte multiplets showed no overall trend in transcriptome quality. **(B)** Ensheathing glia showed a decrease in transcriptome quality in clusters #5 and #7, compared to #3. **(C)** Cortex glia cluster #13 appears to be of a lower quality than #11. **(D)** GO analysis indicating the enriched biological processes in Astrocyte_1 (cluster #0) and Astrocyte_2 (cluster #2). The data underlying this figure can be found in S7 Data and on GitHub at https://github.com/VilFernandesLab/2022_

DrosophilaGlialAtlas.
(TIF)

**S1 Video. 3D projection of newly hatched VNC lateral astrocyte.** 360˚-view of a single laterally positioned astrocyte (*alrm>MCFO*) clone at 0 h after larval hatching.
(MP4)

**S2 Video. 3D projection of newly hatched VNC ventral astrocyte.** 360˚-view of a single ventrally positioned astrocyte (*alrm>MCFO*) clone at 0 h after larval hatching.
(MP4)

**S3 Video. 3D projection of newly hatched VNC dorsal astrocyte.** 360˚-view of a single dorsally positioned astrocyte (*alrm>MCFO*) clone at 0 h after larval hatching.
(MP4)

**S4 Video. 3D projection of an adult epithelial glia (lamina astrocyte).** 360˚-view of a single epithelial glia (*repo>MCFO*) clone in a young adult (1–3 days old).
(MP4)

**S5 Video. 3D projection of an adult long distal medulla astrocyte.** 360˚-view of a single long distal medulla astrocyte (*R86E01>MCFO*) clone in a young adult (1–3 days old).
(MP4)

**S6 Video. 3D projection of an adult short distal medulla astrocyte.** 360˚-view of a single long short medulla astrocyte (*R86E01>MCFO*) clone in a young adult (1–3 days old).
(MP4)

**S7 Video. 3D projection of an adult lateral medulla astrocyte.** 360˚-view of a single lateral medulla astrocyte (*R86E01>MCFO*) clone in a young adult (1–3 days old).
(MP4)

**S8 Video. 3D projection of an adult proximal medulla astrocyte.** 360˚-view of a single proximal medulla astrocyte also known as a chandelier glia (*R86E01>MCFO*) clone in a young adult (1–3 days old).
(MP4)

**S9 Video. 3D projection of an adult lobula-only astrocyte.** 360˚-view of a single lobula-only astrocyte (*R86E01>MCFO*) clone in a young adult (1–3 days old).
(MP4)

**S10 Video. 3D projection of an adult lobula-lobula plate astrocyte.** 360˚-view of a single lobula-lobula-plate astrocyte (*R86E01>MCFO*) clone in a young adult (1–3 days old).
(MP4)

**S11 Video. 3D projection of an adult lobula-only astrocyte.** 360˚-view of a single lobula plate-only astrocyte (*R86E01>MCFO*) clone in a young adult (1–3 days old).
(MP4)

**S1 File. This file is a classification key to distinguish between Embryonic Repo+ glial classes and morphological categories based on association with different brain regions.**
(PDF)

**S2 File. This file is a classification key to distinguish between adult optic lobe glial classes, subclasses, and morphological categories based on association with different brain regions.**
(PDF)

**S3 File. List of specific genotypes and conditions used by figure panel.**
(DOCX)

**S1 Data. List of genes showing enriched expression in midline glia relative to all other embryonic cells.**
(XLSX)

**S2 Data. Lists of genes showing enriched expression in each embryonic *repo+* glial cluster relative to the others.**
(XLSX)

**S3 Data. Lists of genes showing enriched expression in each adult glial cluster relative to the others.**
(XLSX)

**S4 Data. Lists of HCR probe sequences used in this study.**
(XLSX)

**S5 Data. Numerical values used to generate graphs.**
(XLSX)

**S6 Data. Cleaned-up and annotated embryonic glial dataset.**
(ZIP)

**S7 Data. Cleaned-up and annotated (integrated) young adult optic lobe glial dataset.**
(ZIP)

## Acknowledgments

We thank Gaynor Smith, Marc Amoyel, Kelly Monk, Nathan Woodling, Simon Sprecher, and Chris Doe for helpful comments and critiques of the manuscript. Stocks for this study were obtained from the Bloomington Drosophila Stock Center.

## Author Contributions

**Conceptualization:** Sarah D. Ackerman, Vilaiwan M. Fernandes.

**Data curation:** Inês Lago-Baldaia, Maia Cooper, Austin Seroka.

**Formal analysis:** Inês Lago-Baldaia, Maia Cooper, Austin Seroka, Sarah D. Ackerman.

**Funding acquisition:** Stephen W. Wilson, Sarah D. Ackerman, Vilaiwan M. Fernandes.

**Investigation:** Inês Lago-Baldaia, Maia Cooper, Austin Seroka, Sarah D. Ackerman, Vilaiwan M. Fernandes.

**Methodology:** Inês Lago-Baldaia, Maia Cooper, Austin Seroka, Chintan Trivedi, Gareth T. Powell, Sarah D. Ackerman.

**Project administration:** Inês Lago-Baldaia, Vilaiwan M. Fernandes.

**Software:** Inês Lago-Baldaia, Maia Cooper, Austin Seroka.

**Supervision:** Inês Lago-Baldaia, Stephen W. Wilson, Vilaiwan M. Fernandes.

**Validation:** Inês Lago-Baldaia, Maia Cooper, Sarah D. Ackerman.

**Visualization:** Inês Lago-Baldaia, Maia Cooper, Austin Seroka, Sarah D. Ackerman.

**Writing – original draft:** Sarah D. Ackerman, Vilaiwan M. Fernandes.

**Writing – review & editing:** Inês Lago-Baldaia, Maia Cooper, Austin Seroka, Chintan Trivedi, Gareth T. Powell, Stephen W. Wilson, Sarah D. Ackerman, Vilaiwan M. Fernandes.

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
