## [Editor Report · Decision Letter 0]

18 Aug 2023

Dear Dr Fernandes, 

Thank you for submitting your manuscript entitled "A Drosophila glial cell atlas reveals a mismatch between transcriptional diversity and morphological diversity", with portable peer reviews from eLife, for consideration as a Methods and Resource article by PLOS Biology. Your revised manuscript, the reviews from eLife, and your response to reviewers have now been evaluated by the PLOS Biology editorial staff as well as by an academic editor with relevant expertise. I am pleased to say that our Academic Editor is largely satisfied by the changes made, and thinks the manuscript would be suitable for publication at PLOS Biology. However, the Academic Editor has suggested that some very minor textual changes should be considered before publication, and we will likely have some data and policy related editorial requests that will need to be address before formal acceptance. We would therefore like to invite a short revision for your paper. 

**However, before we can send you our 'minor revision' decision, we need you to complete your submission by providing the metadata that is required for full assessment. To this end, please login to Editorial Manager where you will find the paper in the 'Submissions Needing Revisions' folder on your homepage. Please click 'Revise Submission' from the Action Links and complete all additional questions in the submission questionnaire.

Once your full submission is complete, your paper will undergo a series of checks in preparation for peer review. After your manuscript has passed the checks we will send you another email detailing any editorial requests (I anticipate these will be addressable in a very short revision). 

To provide the metadata for your submission, please Login to Editorial Manager (https://www.editorialmanager.com/pbiology) within three working days, i.e. by Aug 23 2023 11:59PM.

As you complete your metadata, please do make sure to fill out the relevant sections on our online system in detail. 

Please also take note of our policies below: 

*PLOS Data Policy*

*Blot and Gel Data Policy*

Kind regards,

Luke

Lucas Smith, Ph.D.

Senior Editor

PLOS Biology

lsmith@plos.org

---

## [Editor Report · Decision Letter 1]

24 Aug 2023

Dear Dr Fernandes,

Thank you for providing the metadata related to your manuscript "A Drosophila glial cell atlas reveals a mismatch between transcriptional diversity and morphological diversity", which is currently under consideration as a Methods and Resources article at PLOS Biology, submitted with portable peer reviews from multiple rounds of review at eLife. As mentioned in my last email, your revised manuscript, and your response to reviewers from eLife, was previously assessed by an Academic Editor who felt that the study was suitable for publication at PLOS Biology with only minor revisions. Below I detail some editorial requests that need to be addressed before we can accept your study. 

From the review history, we see that reviewers 2 and 3 from eLife were satisfied by the revision but reviewer 1 had some lingering concerns about the specificity of CG5080-Gal4 as a marker of perineurial glia. We see that that reviewer 1 argued that findings from an earlier report (Beckervordersandforth et al., 2008) suggests that CG5080 is coexpressed with the SPG marker moody - and reviewer 1 felt that more work was needed to demonstrate CG5080 as a marker of PGs.

We understand that you disagree with this interpretation, noting that CG5080 and moody co-expression was actually not directly shown in the previous work. Our Academic Editor largely agrees with your interpretation of the previous study, and s/he thinks the the current data provided is adequate to support your conclusions about perineurial glial cell types. Therefore, based on this assessment, we would not require that this conclusion be removed or experimentally strengthened before publication. 

However, as the disagreement comes down to interpretation, and as other readers may have similar questions as reviewer 1, we suggest that you consider adding further discussion regarding reviewer 1's comments to the manuscript. We think it would be helpful to outline in a few sentences detailing why there is a perceived discrepancy. In particular, the the Academic Editor suggests you could expand on this comment from the results "Thus, based on morphology, tiling properties, size and position, these data argue that moody-Gal4 labels the subperineurial glial class, which can be further subdivided into surface-only, surface- and channel-associated, and channel-only morphological categories. By contrast, CG5080- Gal4 labels the perineurial glial class, which can be further subdivided into surface-only and channel associated morphological categories."

**IMPORTANT: As you address this last comment from the Academic Editor, we also ask that you attend to the following editorial requests: 

1) TITLE: We would like to suggest a minor tweak to streamline the title. If you agree, we suggest it be changed to: "A Drosophila glial cell atlas reveals a mismatch between transcriptional and morphological diversity"

2) DATA AND CODE: Thank you for providing the underlying data and code related to your study as depositions and a supplemental file. I have a couple minor requests related to these datasets: 

a. Please update the github link https://github.com/AustinSeroka/2022_stage17_glia, to include a readme file, detailing what the various codes are and where they were applied in the manuscript.

b. Please add a sentence to every figure legend (including supplemental) indicating where the underling data can be found. For example, you can add the sentence "the data underlying this figure can be found at ___" (and then reference the relevant repository/supplemental file)

3) DATA NOT SHOWN: Please note that per journal policy, we do not allow the mention of "data not shown", "personal communication", "manuscript in preparation" or other references to data that is not publicly available or contained within this manuscript. I detected two instances of this on line 262 and 273. Please either remove mention of these data or provide figures presenting the results and the data underlying the figure(s).

We are pleased to invite you to address these editorial requests in a revision that we anticipate will not take very long. We expect to receive your revised manuscript within two weeks. 

*Published Peer Review History*

*Press*

Sincerely,

Luke

Lucas Smith, Ph.D.

Senior Editor,

lsmith@plos.org,

PLOS Biology

---

## [Editor Report · Decision Letter 2]

6 Sep 2023

Dear Dr Fernandes,

Thank you for your patience while we considered your revised manuscript "A Drosophila glial cell atlas reveals a mismatch between transcriptional and morphological diversity" for publication as a Methods and Resources at PLOS Biology. This revised version of your manuscript has been evaluated by the PLOS Biology editors and the Academic Editor.

Based on our Academic Editor's assessment of your revision, we are likely to accept this manuscript for publication, but we would like you to address one remaining issue about the R56F03-Gal4 data. We had mentioned in the previous email that per journal policy, we do not allow the mention of "data not shown", "personal communication", "manuscript in preparation" or other references to data that is not publicly available or contained within this manuscript. 

We note that you have deleted the "data not shown" statement in line 277 but still mention the data. Please either include the data in the manuscript or remove the corresponding reference to this dataset. I didn't see any other references to this dataset apart from the methods section, so you could simply delete the sentence if you do not discuss this dataset anywhere else. 

We expect to receive your revised manuscript within one week. 

*Published Peer Review History*

*Press*

Sincerely,

Christian Schnell (on behalf of Lucas who will be continue handling your manuscript when he is back in the office next week)

Christian Schnell, Ph.D.

Senior Editor

cschnell@plos.org

PLOS Biology

Lucas Smith, Ph.D.

Senior Editor,

lsmith@plos.org,

PLOS Biology

---

## [Editor Report · Decision Letter 3]

8 Sep 2023

Dear Dr Fernandes,

Thank you for the submission of your revised Methods and Resources "A Drosophila glial cell atlas reveals a mismatch between transcriptional and morphological diversity" for publication in PLOS Biology. On behalf of my colleagues and the Academic Editor, Cody Smith, I am pleased to say that we can in principle accept your manuscript for publication, provided you address any remaining formatting and reporting issues. These will be detailed in an email you should receive within 2-3 business days from our colleagues in the journal operations team; no action is required from you until then. Please note that we will not be able to formally accept your manuscript and schedule it for publication until you have completed any requested changes.

PRESS

Sincerely, 

Christian Schnell (on behalf of Lucas)

Senior Editor

PLOS Biology

cschnell@plos.org

Lucas Smith, Ph.D., Ph.D.

Senior Editor

PLOS Biology

lsmith@plos.org